# Local shape descriptors for neuron segmentation

**Arlo Sheridan[1,2], Tri M. Nguyen ⬡[3], Diptodip Deb[1], Wei-Chung Allen Lee ⬡[4], Stephan Saalfeld ⬡[1], Srinivas C. Turaga ⬡[1], Uri Manor ⬡[2] & Jan Funke ⬡[1]✉**

We present an auxiliary learning task for the problem of neuron segmentation in electron microscopy volumes. The auxiliary task consists of the prediction of local shape descriptors (LSDs), which we combine with conventional voxel-wise direct neighbor affinities for neuron boundary detection. The shape descriptors capture local statistics about the neuron to be segmented, such as diameter, elongation, and direction. On a study comparing several existing methods across various specimen, imaging techniques, and resolutions, auxiliary learning of LSDs consistently increases segmentation accuracy of affinity-based methods over a range of metrics. Furthermore, the addition of LSDs promotes affinity-based segmentation methods to be on par with the current state of the art for neuron segmentation (flood-filling networks), while being two orders of magnitudes more efficient—a critical requirement for the processing of future petabyte-sized datasets.

The goal of connectomics is the reconstruction and interpretation of neural circuits at synaptic resolution. These wiring diagrams provide insight into the inner mechanisms underlying behavior and help drive future theoretical experiments[1–4]. Additionally, the generation of connectomes complements existing techniques such as calcium imaging and electrophysiology where the resolution is often not sufficient to parse the circuitry in detail[5,6].

Currently, only electron microscopy (EM) allows imaging of neural tissue at a resolution sufficient to resolve individual synapses and fine neural processes. Two popular methods for imaging these volumes are serial block-face scanning EM (SBFSEM) and focused ion beam scanning EM (FIB-SEM). While the former technique is faster and generates high lateral resolution, it results in lower axial resolution owing to section slicing. The latter method produces isotropic resolution by etching the face of the volume with a focused ion beam before imaging. However, this method is slower than serial section approaches. Previous work[7] provides a thorough overview of these imaging approaches and others, including serial section transmission EM (ssTEM) and automated tape-collecting ultramicrotome scanning EM (ATUM-SEM). All methods have been used to generate invaluable datasets for the connectomics community[8–15].

Depending on the specimen and the circuit of interest, current EM acquisitions produce datasets ranging from several hundred terabytes to petabytes. For instance, the raw data of a full adult fruit fly brain (FAFB) comprises ~50 teravoxels of neuropil[16]. Even sub-volumes taken from vertebrate brains, which do not contain brain-spanning circuits, result in massive amounts of data. One example is a region taken from a zebrafinch brain containing ~10^6 μm^3 (~663 gigavoxels) of raw data[17]. A larger volume of mouse visual cortex was recently imaged, comprising ~3 × 10^6 μm^3 (~6,614 gigavoxels)[11–13,18,19]. A 1.4 petabyte volume taken from human cortex further demonstrates the rapid advances in massive dataset acquisition[20]. To reconstruct circuits in a full mouse brain, however, it will require the acquisition of around 1 exabyte of raw data (1,000,000 terabytes)[21].

With datasets of this magnitude, purely manual reconstruction of connectomes is infeasible. On average, manual tracing in a mouse tissue takes ~1–2 h per millimeter[2,22]. Larval tissue averages ~13.7 h per millimeter[1], which is comparable to reported tracing speeds of 4–13 hours per millimeter in the *Drosophila* dataset FAFB[10,29], owing to the challenging nature of invertebrate neuropil. Even the small brain of a *Drosophila* contains an estimated 100,000 neurons, which would require ~125 years of manual effort to trace each neuron to completion.

[1]HHMI Janelia, Ashburn, VA, USA. [2]Waitt Advanced Biophotonics Center, Salk Institute for Biological Studies, La Jolla, CA, USA. [3]Department of Neurobiology, Harvard Medical School, Boston, MA, USA. [4]F.M. Kirby Neurobiology Center, Boston Children's Hospital, Harvard Medical School, Boston, MA, USA. ✉e-mail: funkej@janelia.hhmi.org

Consequently, automatic methods for the reconstruction of neurons and identification of synapses have been developed. Over the past decade, methods targeting relatively small volumes have pioneered the reconstruction of neurons[23,24] and synapses[25,26]. More recently, these efforts have been improved to tackle the challenges of large datasets for neurons[11,27–29], synaptic clefts[16] and synaptic partners[30,31]. With the help of an automatic neuron segmentation method, neuron tracing times decreased by a factor of 5.4 - 11.6[29], effectively trading compute time for human tracing time.

However, given the daunting sizes of current and future EM datasets, limits on available compute time become a concern. Future algorithms do not only need to be more accurate to further decrease manual tracing time but also computationally more efficient to be able to process large datasets in the first place. Consider the computational time required by the current state of the art, flood-filling network (FFN): assuming linear scalability and the availability of 1,000 contemporary GPUs (or equivalent hardware), the processing of a complete mouse brain would take about 226 years. This example alone goes to show that the objective for future method development should be the minimization of the total time spent to obtain a connectome, including computation and manual tracing. Therefore, automatic methods for connectomics need to be fast, scalable (that is, trivially parallelizable) and accurate.

To address this, we developed local shape descriptors (LSDs) as an auxiliary learning task for boundary detection. The motivation behind LSDs (distinct from previous shape descriptors[32]) is to provide an auxiliary learning task that improves boundary prediction by learning statistics describing the local shape of the object close to the boundary. Previous work demonstrated a similar technique to yield superior results over boundary prediction alone[33]. Here, we extend on this idea by predicting for every voxel not just affinities values to neighboring voxels, but also statistics extracted from the object under the voxel aggregated over a local window, specifically (1) the volume, (2) the voxel-relative center of mass and (3) pairwise coordinate correlations (Figs. 1 and 2). We demonstrate that when using LSDs as an auxiliary learning task, segmentation results are competitive with the current state of the art, albeit two orders of magnitude more efficient to compute. We hope that this technique will allow laboratories to generate accurate neuron segmentations for their connectomics research using standard compute infrastructure.

## Results

Here, we present experimental results of the LSDs for neuron segmentation. We compare the accuracy of LSD segmentations against several alternative methods for affinity prediction and FFN on three large and diverse datasets we refer to as ZEBRAFINCH[17], HEMI-BRAIN[14] and FIB-25[8]. Furthermore, we compare the computational efficiency of different methods and analyze the relationship between different error metrics for neuron segmentations.

### Investigated methods

For each dataset we investigated seven methods:

- Direct neighbor affinities (BASELINE): baseline network with a single voxel affinity neighborhood and mean squared error (MSE) loss[23]. We trained a three-dimensional (3D) U-NET to predict affinities.
- Long-range affinities (LR): same approach as the BASELINE network but uses an extended affinity neighborhood with three extra neighbors per direction[24]. The extended neighborhood functions as an auxiliary learning task to improve the direct neighbor affinities.
- MALIS loss (MALIS): same approach as the BASELINE network, but using MALIS loss[28] instead of plain mean squared error (MSE).

- Flood-filling networks (FFN): a single segmentation per investigated dataset from the current state of the art approach[27].
- Multitask LSDs (MTLSD): a network to predict both LSDs and direct neighbor affinities in a single pass, (Fig. 1e). Similar to LR, the LSDs act as an auxiliary learning task for the direct neighbor affinities.
- Auto-context LSDs (ACLSD): an auto-context setup, where LSDs were predicted from one network and then used as input to a second network in which affinities were predicted.
- Auto-context LSDs with raw (ACRLSD): same approach as ACLSD, but the second network also receives the raw data as input in addition to the LSDs generated by the first network.

All network architectures for the ZEBRAFINCH and FIB-SEM volumes are described in detail in the Supplementary Note. To fairly evaluate accuracy as a function of only the used segmentation method, we made sure to hold other contributing factors constant. We trained each affinity-based network with the same pipeline (for example, data augmentations and optimizer) and same hyper-parameters for each dataset. We also used the masks to restrict segmentation and evaluation to dense neuropil. These are the same masks used by FFN, thus comparing pure neuron segmentation performance of each method.

Since proofreading of segmentation errors is currently the main bottleneck in obtaining a connectome[14], the metrics to assess neuron segmentation quality should ideally reflect the time needed for proofreading. This requirement is not easily met, since it depends on the tools and strategies used in a proofreading workflow. Currently used metrics aim to correlate scores with the time needed to correct errors on the basis of assumptions about the gravity of certain types of errors. A common assumption is that false merges take substantially more time to correct than false splits, although next-generation proofreading tools challenge this conception[34–36].

In this study, we report neuron segmentation quality with two established metrics: variation of information (VOI) and expected run length (ERL). In addition to those metrics, we propose the min-cut metric (MCM), designed to measure the number of graph edit operations needed to perform in a hypothetical proofreading tool (Supplementary Note and Extended Data Fig. 1).

### Segmentation accuracy in a ZEBRAFINCH SBFSEM dataset

A volume from neural tissue of a songbird was the largest dataset used in this study[17,27]. This volume consists of a ~$10^6$ μm³ region of a zebrafinch brain, imaged with SBFSEM at a resolution of 9 × 9 × 20 nm ($x × y × z$) (Fig. 3 and Supplementary Note). For our experiments, we used a slightly smaller region completely contained inside the raw data with edge lengths of 87.3, 83.7 and 106 μm, respectively ($x$, $y$ and $z$). We refer to this region as the BENCHMARK region of interest (ROI).

For each affinity-based network described above, we used 33 volumes containing a total of ~200 μm³ (~6 μm³ average per volume) of labeled data[27] for training. We then ran prediction on the BENCHMARK ROI, using a block-wise processing scheme.

Using the resulting affinities, we generated two sets of supervoxels: one without any masking and one constrained to neuropil using a mask[27]. Additionally, we filtered supervoxels in regions in which the average affinites were lower than a predefined value (for example, glia). Supervoxels were agglomerated using one of two merge functions[28], to produce the region adjacency graphs used for evaluation.

We then produced segmentations for ROIs of varying size centered in the BENCHMARK ROI, to assess how segmentation measures scale with the volume size. In total, we cropped ten cubic ROIs ranging from ~11 μm to ~76 μm edge lengths, in addition to the whole BENCHMARK ROI. We will refer to the respective ROIs by their edge lengths. For each affinity-based network, in each ROI, we created segmentations for a range of agglomeration thresholds (resulting in a sequence of segmentations ranging from over- to undersegmentation). Additionally,

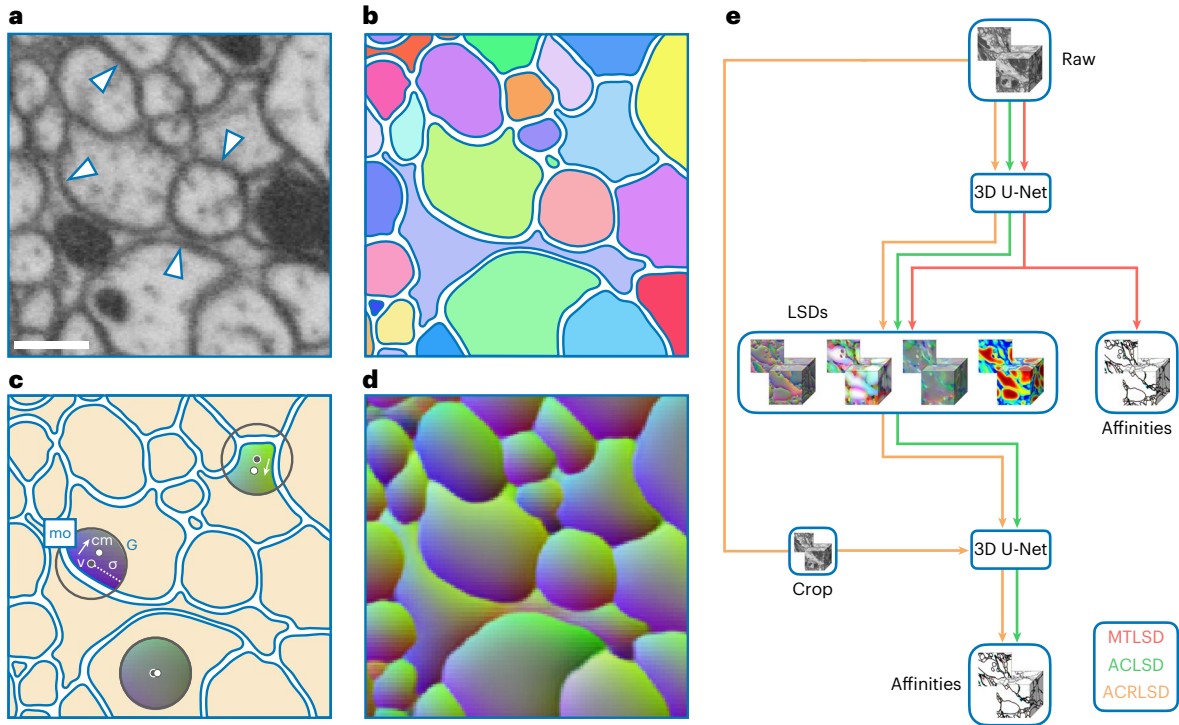

**Fig. 1 | LSD and network architecture overview. a**, EM data imaged with FIB-SEM at 8 nm isotropic resolution (FIB-25 dataset[8]). Arrows point to example individual neuron plasma membranes. Dark blobs are mitochondria. Scale bar, 300 nm. **b**, Label colors correspond to unique neurons. **c**, LSD mean offset schematic. A Gaussian (G) with fixed sigma ($\sigma$) is centered at voxel (v). The Gaussian is then intersected with the underlying label (colored region) and the center of mass of the intersection (cm) is computed. The mean offset (mo) between the given voxel and center of mass is calulated (among several other statistics), resulting in the first three components of the LSD for voxel (v). **d**, Predicted mean offset component of LSDs (LSD[0:3]) for all voxels. A smooth gradient is maintained within objects while sharp contrasts are observed across boundaries. Three-dimensional vectors are RGB color encoded. **e**, Network architectures used. The ten-dimensional LSD embedding is used as an auxiliary learning task for improving affinities. In a multitask approach (MTLSD), LSDs and affinities are directly learnt. In an auto-context approach, the predicted LSDs are used as input to a second network to generate affinities both without raw data (ACLSD) and with raw data (ACRLSD).

we cropped the provided FFN segmentation accordingly and relabeled connected components.

We used a set of 50 manually ground-truthed skeletons[27], comprising 97 mm, for evaluation. For each network we assessed VOI and ERL on each ROI. For affinity-based methods we also computed the MCM on the 11, 18 and 25 µm ROIs. Additionally, we used 12 validation skeletons consisting of 13.5 mm to determine the optimal thresholds for each network on the BENCHMARK ROI (Supplementary Note).

We find that LSDs are useful for improving the accuracy of direct neighbor affinities and subsequently the resulting segmentations (Fig. 4). Specifically, LSD-based methods consistently outperform other affinity-based methods over a range of ROIs, whether used in a multitask (MTLSD) or auto-context (ACLSD and ACRLSD) architecture (Fig. 4a and Supplementary Note). In terms of segmentation accuracy according to VOI, the best auto-context network (ACRLSD) performs on par with FFN (Fig. 4a).

We find that the ranking of methods depends on the size of the evaluation ROI. Even for monotonic metrics like VOI, we see that performance on the smallest ROIS (up to 54 µm) does not extrapolate to the performance on larger datasets.

We also investigated how ERL varies over different ROI sizes. To this end, we cropped the skeleton ground-truth to the respective ROIs and relabeled connected components (as we did for the VOI evaluation). However, the resulting fragmentation of skeletons heavily impacts ERL scores: ERL cannot exceed the average length of skeletons, and thus the addition of shorter skeleton fragments can result in a decrease of ERL, even in the absence of errors. ERL measures do not progress monotonically over ROI sizes and absolute values are likely not comparable across different dataset sizes (Fig. 4b). In addition, the ranking of methods for a given ROI size varies substantially over different ROI sizes. The discrepancy between the computed ERL and maximum possible ERL (or the ground-truth ERL) further emphasizes this point (Supplementary Note).

Furthermore, the ERL metric is by design very sensitive to merge errors, as it considers a whole neuron to be segmented incorrectly if it was merged with even only a small fragment from another neuron. Thus, merge errors contribute disproportionally to the ERL computation. In addition, the contribution depends on the sizes of the merged segments. Merging a small fragment of one neuron into an otherwise correctly reconstructed large neuron will have a larger negative impact on the ERL than merging two small fragments from different neurons, although the effort needed to resolve that error is likely the same. We observe that this property leads to erratic scores across different volume sizes (Fig. 4b and Supplementary Note) that no longer reflect the amount of time needed to proofread the resulting segmentation. The sensitivity to merge errors also contributes to the observed differences between the ERL scores of the LSD-based methods and FFN (Fig. 4b). Although ACRLSD has a lower total VOI than FFN (2.239 versus 2.256), ACRLSD has a higher merge rate than FFN with a (ACRLSD VOI merge score of 1.436 versus FFN VOI merge score of 1.118), resulting in substantially different ERL scores of 13.5 µm for ACRLSD and 16.7 µm for FFN (Supplementary Note).

The high variability between metrics and ROI sizes prompted us to develop a metric that aims to measure proofreading effort. We developed MCM to count the number of interactions needed to split and merge neurons to correctly segment the ground-truth skeletons, assuming that a min-cut-based split tool is available. Owing to the

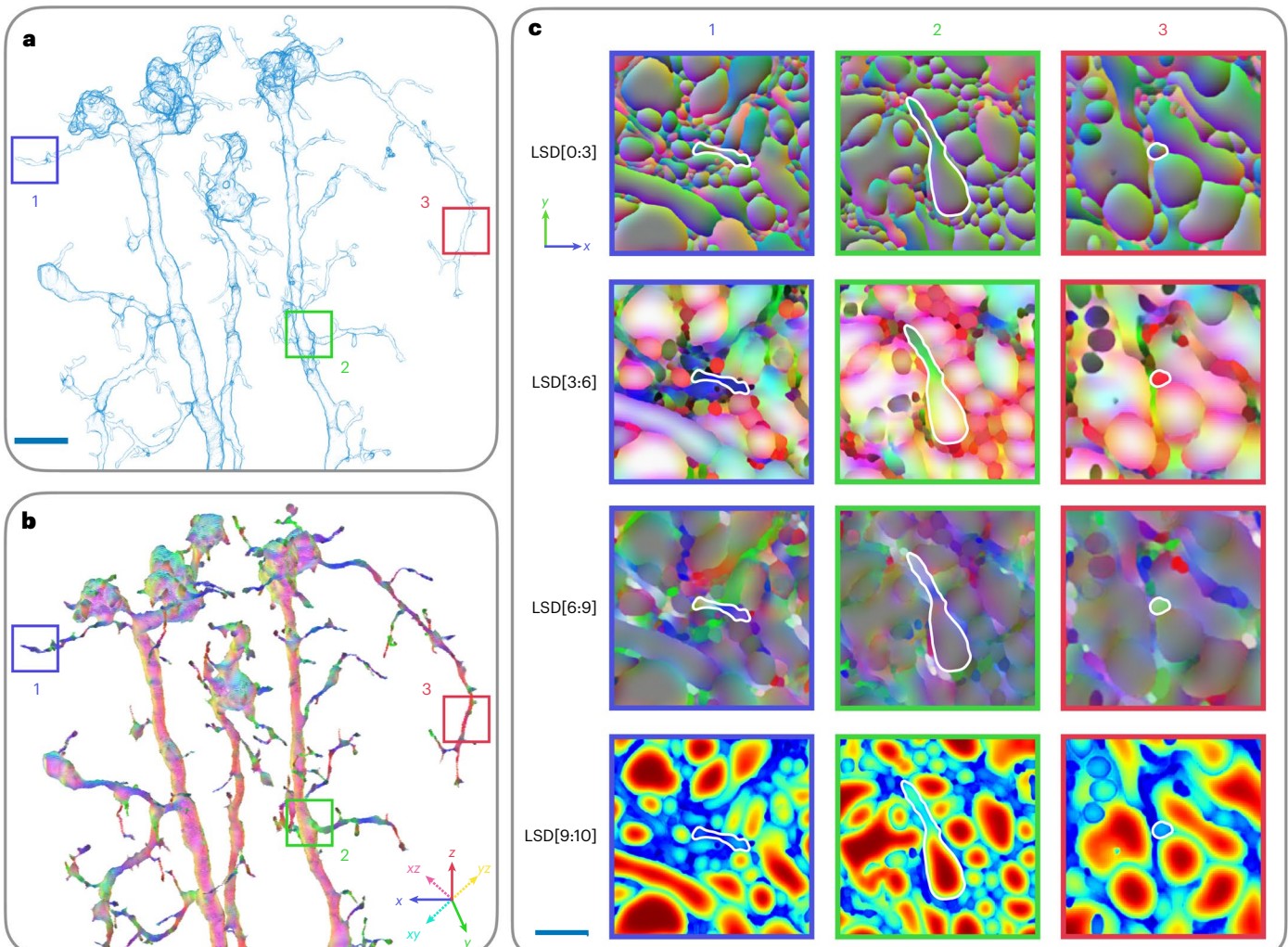

**Fig. 2 | Visualization of LSD components. a**, Surface mesh of a segmented neuron from FIB-SEM data (FIB-25[8] dataset). Scale bar, 1 μm. **b**, RGB mapping of LSD components 3, 4 and 5. Neural processes are colored with respect to the directions they travel. Intermediate directions are mapped accordingly (see|Cartesian coordinate inset). **c**, LSD predictions in corresponding two-dimensional slices to the three boxes shown in **a,b**; neuron highlighted in white. Columns signify neuron orientation (blue, lateral movement; green, vertical movement; red, through-plane movement). Rows correspond to components of the LSDs. First row, mean offset; second and third rows, covariance of coordinates (LSD[3:6] for the diagonal entries, LSD[6:9] for the off-diagonals), second row shows mapping seen in **b**; last row, size (number of voxels inside intersected Gaussian). Scale bar, 250 nm.

computational cost associated with MCM (stemming from repeated min-cuts in large fragment graphs), we limited its computation to the three smallest investigated ROIs in this dataset. As expected, we observe a linear increase in MCM with ROI size across different methods (Fig. 4c). Furthermore, we see that MCM and VOI mostly agree on the ranking of methods (Fig. 4d and Supplementary Note), which suggests that VOI should be preferred to compare segmentation quality in the context of a proofreading workflow that allows annotators to split false merges using a min-cut on the fragment graph. Since the MCM requires a supervoxel graph, it was not possible to compute on the single FFN segmentation provided.

Binary masks are commonly used to limit neuron segmentation to dense neuropil and exclude confounding structures like glia cells. Recent approaches to processing large volumes have incorporated tissue masking at various points in the pipeline[11,14,27,29] to prevent errors in areas that were underrepresented in the training data. Our results confirm the importance of masking. We used a neuropil mask which excluded cell bodies, blood vessels, myelin and out-of-sample background voxels (Supplementary Note). Across all investigated methods, the accuracy degraded substantially on larger ROIs when processed without masking (Fig. 4f, Supplementary Note).

### Segmentation accuracy in *Drosophila* FIB-SEM datasets

We also evaluated all architectures on two *Drosophila* datasets imaged with FIB-SEM at 8 nm resolution (Figs. 3 and 5 and Supplementary Note) and found results to generally be consistent with the ZEBRAFINCH (Extended Data Fig. 2 and Supplementary Note). Since the majority of large connectomics datasets are imaged with ssTEM from mammalian tissue, we conducted an extended experiment to evaluate several networks on small volumes of mouse visual cortex[19]. We generally find consistent results to the other datasets; LSD networks outperform baseline methods most noticeably when used in an auto-context setup (Supplementary Note). While the available volumes were likely too small to directly infer performance on larger data, we expect LSDs to benefit from the same data-specific processing strategies that other methods routinely use.

### Computational efficiency of LSD-based networks

In addition to being accurate, it is important for neuron segmentation methods to be fast and computationally inexpensive. As described in the introduction, the acquisition size of datasets is growing rapidly and approaches should therefore aim to complement this trajectory. Since LSDs only add a few extra feature maps to the output of the U-NET,

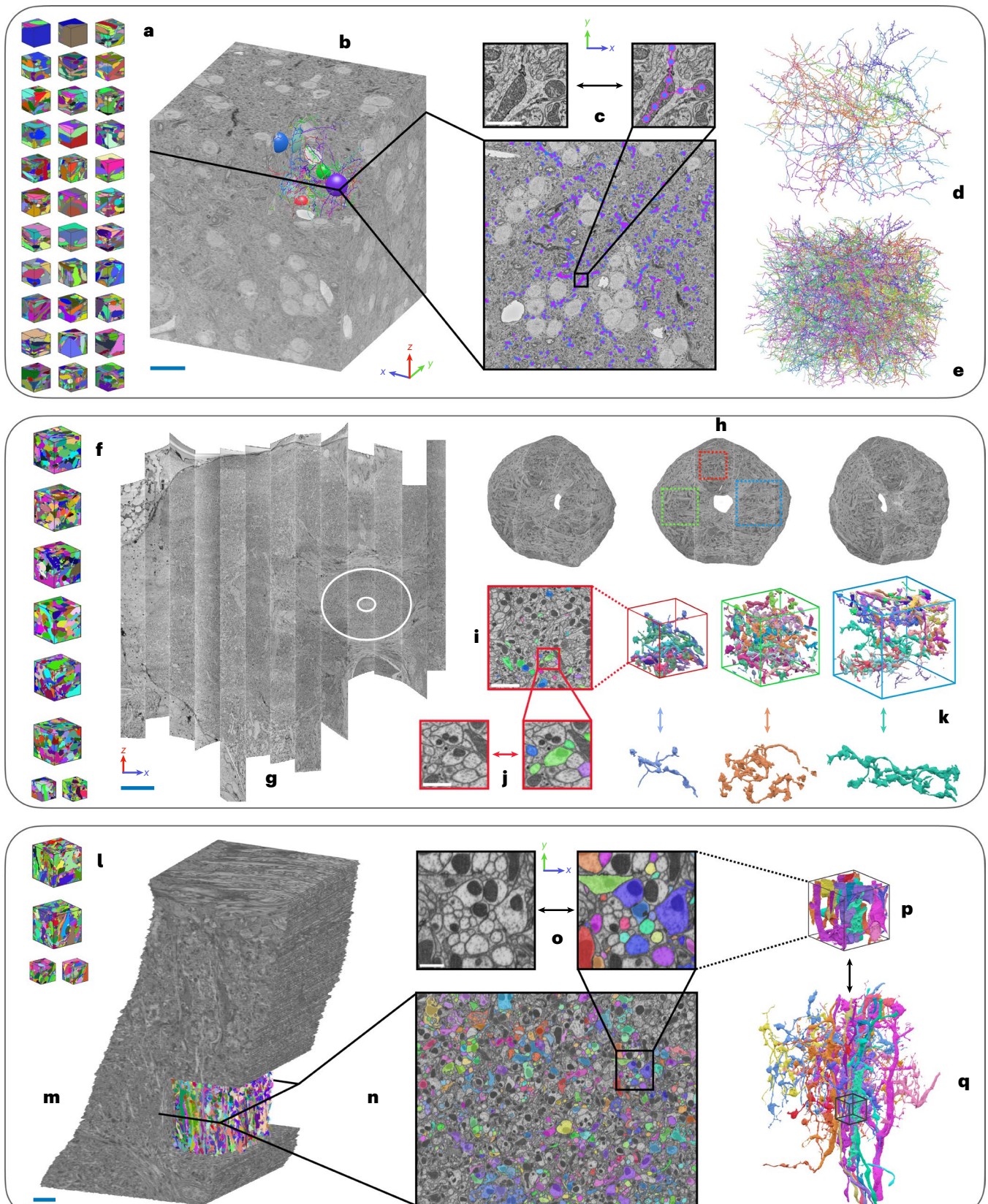

**Fig. 3 | Overview of datasets. a**, ZEBRAFINCH dataset[17]. Thirty-three gound truth volumes were used for training. **b**, Full raw dataset. Scale bar, 15 µm. **c**, Single section shows ground-truth skeletons. Zoom-in scale bar, 500 nm. **d**, Validation skeletons (*n* = 12). **e**, Testing skeletons (*n* = 50). **f**, HEMI-BRAIN dataset[14]. Eight ground-truth volumes were used for training. **g**, Full HEMI-BRAIN volume. Scale bar, 30 µm. Experiments were restricted to ELLIPSOID BODY (circled region). **h**, Volumes used for testing. **i**, Example sparse ground-truth testing data. Scale bar, 2.5 µm. **j**, Zoom-in scale bar, 800 nm. **k**, Example 3D renderings of selected neurons. **l**, FIB-25 dataset[8]. Four ground-truth volumes were used for training. **m**, Full volume with cutout showing testing region. Scale bar, 5 µm. **n**, Cross section with sparsely labeled testing ground-truth. **o**, Zoom-in scale bar, 750 nm. **p**, Sub-volume corresponding to zoomed-in plane. **q**, Subset of full ROI testing neurons. Small volume shown for context.

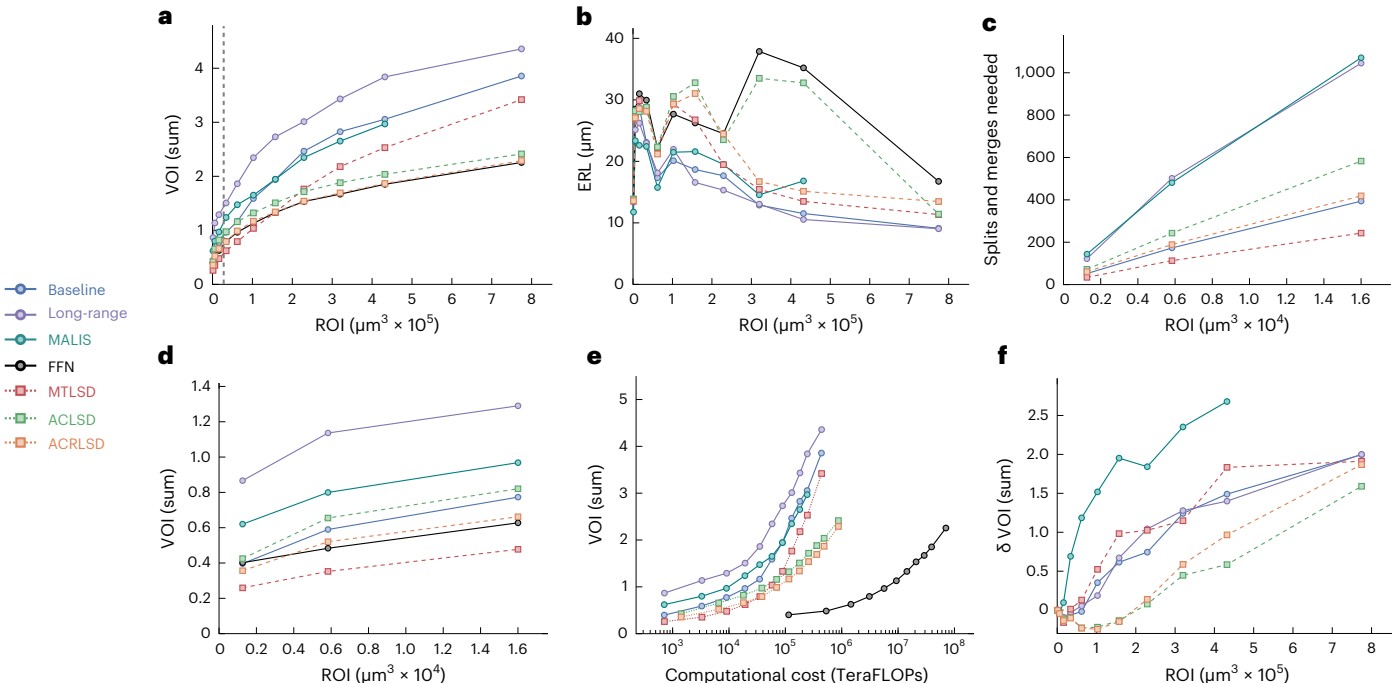

**Fig. 4 | Quantitative results on ZEBRAFINCH dataset.** Points in plots correspond to optimal thresholds from validation set. Each point represents an ROI. For VOI and MCM, lower scores are better; for ERL, higher scores are better. **a**, VOI sum versus ROI size (μm³). **b**, ERL (nanometers) versus ROI size. **c,d**, MCM sum and VOI sum versus ROI size (first three ROIs), respectively. Dashed line in **a** corresponds to ROIS shown in **c,d. e**, TERAFLOPS versus VOI sum across ROIs (as in **a,b**). **f**, Mask δ VOI sum versus ROI.

there is almost no difference in computational efficiency compared to BASELINE affinities. LSD-based methods can therefore be parallelized in the same manner as affinities, making them a good candidate for the processing of very large datasets or environments with limited computing resources.

In our experiments, we computed prediction and segmentation of affinity-based methods in a block-wise fashion, allowing parallel processing across many workers (Fig. 6). This allowed for efficient segmentation following prediction (Supplementary Note).

When considering computational costs in terms of floating point operations (FLOPS), we find that the ACRLSD network (the computationally most expensive of all LSD architectures) is two orders of magnitude more efficient than FFN, while producing a segmentation of comparable quality (Fig. 4e). For this comparison, we computed FLOPS of all affinity-based methods during prediction (Supplementary Note). For FFN, we used the numbers reported in ref. 27, limited to the forward and backward passes of the network, that is, the equivalent of the prediction pass for affinity-based methods. We limit the computational cost analysis to GPU operations, since FLOP estimates on CPUs are unreliable and the overall throughput is dominated by GPU operations. We therefore only consider inference costs for all affinity-based networks, since agglomeration is a post-processing step done on the CPU. To keep the comparison to FFN fair, we do not count FLOPS during FFN agglomeration, although it involves a substantial amount of GPU operations. Generally, affinity-based methods are more computationally efficient than FFN by two orders of magnitude when considering FLOPS (Supplementary Note).

FFN throughput can be improved by a factor of five using a 'coarse-to-fine' approach in which multiple models are trained at different scales and then the segmentations are merged using an oversegmentation-consensus[27]. Data at the highest resolution is often not necessary to resolve large objects (such as axonal tracks and boutons). Since we computed every affinity-based method on the highest resolution, further speed ups are likely achievable by adapting these methods to run at different resolutions and are a logical next step.

## Discussion

The main contribution of this work is the introduction of LSDs as an auxiliary learning task for neuron segmentation. All methods, datasets and results are publicly available (https://github.com/funkelab/lsd), which we hope will be a useful starting point for further extensions and a benchmark to evaluate future approaches in a comparable manner.

Auxiliary learning tasks have been shown to improve network performance across different applications. One possible explanation for why auxiliary learning is also helpful for the prediction of neuron boundaries is that the additional task incentivizes the network to consider higher-level features. Predicting LSDs is likely harder than boundaries, since additional local structure of the object has to be considered. Merely detecting an oriented, dark sheet (for example, plasma membranes) is not sufficient; statistics of the whole neural process have to be taken into account. Those statistics rely on features that are not restricted to the boundary in question. Therefore, the network is forced to make use of more information in its receptive field than is necessary for boundary prediction alone. This, in turn, increases robustness to local ambiguities and noise for the prediction of LSDs. As a welcome side effect, it seems that the network learns to correlate boundary prediction with LSD prediction, which explains why the boundary prediction benefits from using the LSDs as an auxiliary objective.

In an auto-context learning strategy, the quality of a prediction is refined by using a cascade of predictors[37]. We loosely adapted this idea when designing our networks (ACLSD and ACRLSD) and found that it helped to improve segmentations across all datasets. We tested if this increase in accuracy was consistent when using affinities as the input to the second network (that is, a BASELINE auto-context approach, ACBASELINE) and found that it made no substantial improvements to the BASELINE network (Extended Data Fig. 3). We hypothesize that predicting affinities from affinities is too similar to predicting affinities from raw EM data. Specifically, we suspect that the ACBASELINE network simply copies data in the second pass rather than learning anything new. Easy solutions, such as looking for features like oriented

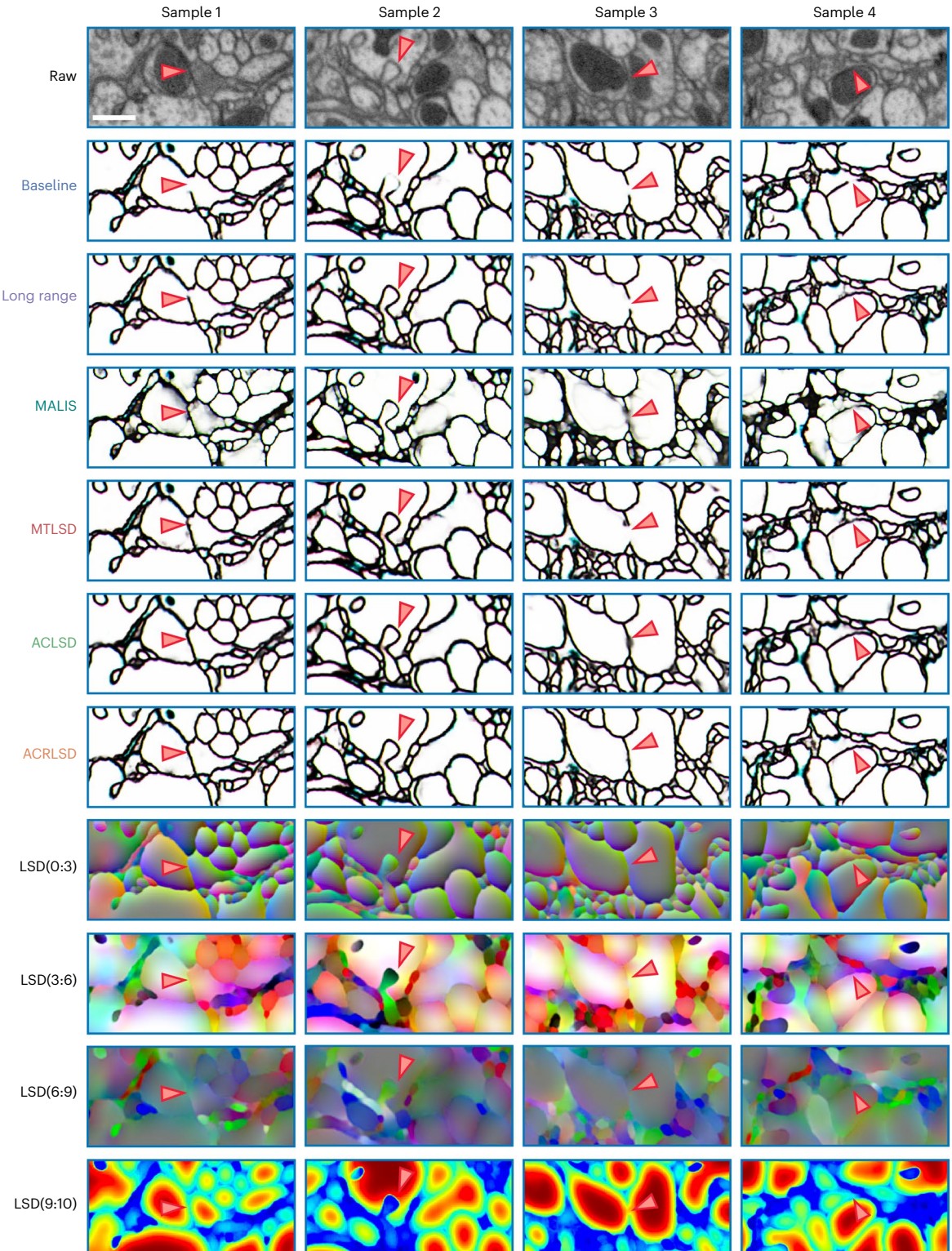

**Fig. 5 | Qualitative results on FIB-25 dataset.** Top row shows raw data. Arrows correspond to ambiguous plasma membranes, which might lead to merge errors. Scale bar, 500 nm.

bars, already produce relatively accurate boundaries in the first pass. Consequently, there is little incentive for the network to change course in the second pass. Translating from LSDs to affinities, on the other hand, is a comparatively different task, which forces the network to incorporate the features from the LSDs in the second pass. The subsequent boundary predictions seem to benefit from this.

One of the challenges of deep learning is to find representative testing data and metrics to infer production performance. This is especially challenging for neuron segmentation, considering the diversity of neural ultrastructure and morphology found in EM volumes. While challenges like CREMI and SNEMI3D (http://brainiac2.mit.edu/SNEMI3D) make an effort to include representative training

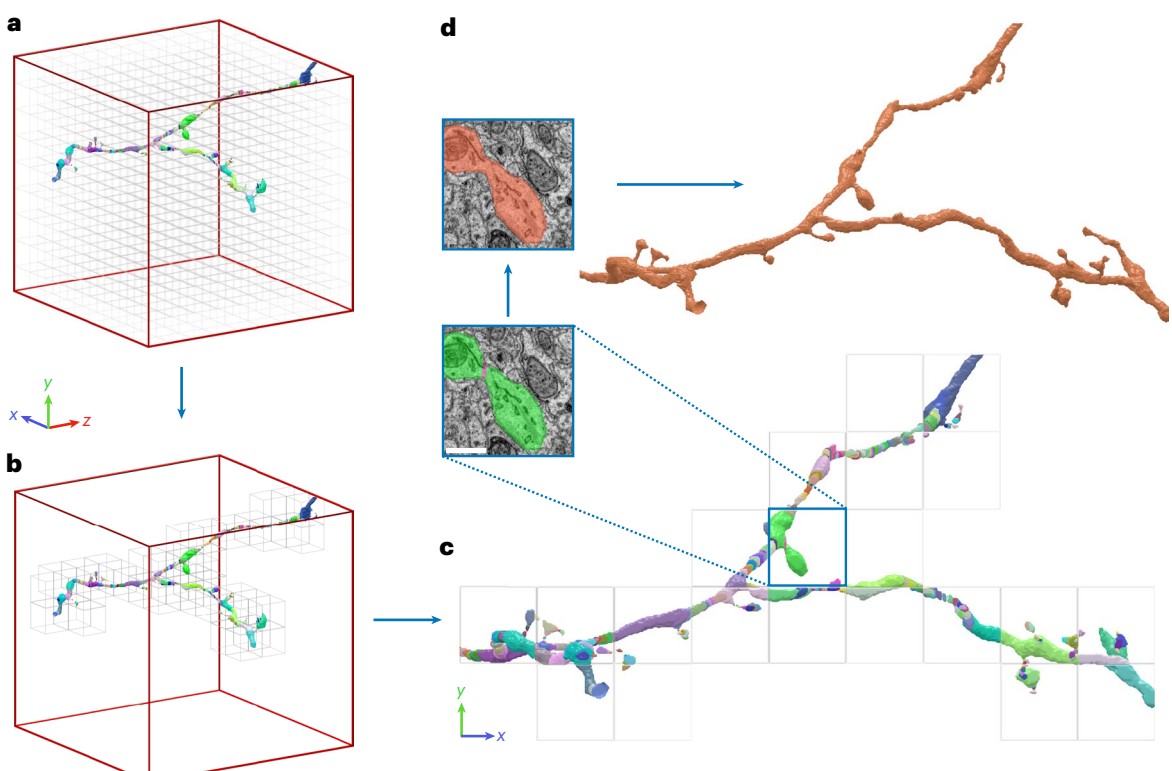

**Fig. 6 | Overview of block-wise processing scheme. a**, Example 32-μm ROI showing total block grid. **b**, Required blocks to process example neuron. Scale bar, ~6 μm. **c**, Corresponding orthographic view highlights supervoxels generated during watershed. Block size, 3.6 μm. Inset shows respective raw data inside single block. Scale bar, ~1 μm. **d**, Supervoxels are then agglomerated to obtain a resulting segment. Note, while this example shows processing of a single neuron, in reality all neurons are processed simultaneously.

and testing data, the implications for model performance on larger datasets are not straightforward. Our results suggest that testing on small volumes provides limited insight into the quality of a method when applied to larger volumes. For example, the total volume of the three CREMI testing datasets (~1,056 μm³) is still less than the smallest ZEBRAFINCH (~1,260 μm³) and HEMI-BRAIN (~1,643 μm³) ROIs. In this context, it seems difficult to declare a clear 'winner' when it comes to neuron segmentation accuracy. Dataset sizes and the choice of evaluation metrics greatly influence which method is considered successful.

## Online content

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

## Methods

### LSDs

Intuitively, the LSD components encourage the neural network to make use of its entire field of view (FOV) to reach a decision about the presence or absence of a boundary in the center of the field of view. Trained on a boundary prediction task alone (that is, pure affinity-based methods), a neural network might focus only on a few center voxels to detect membranes and achieve high accuracy during training, especially if trained using a voxel-wise loss. However, this strategy might fail in rare cases where boundary evidence is ambiguous. Those rare cases contribute little to the training loss, but given the large size of datasets in connectomics, those cases still result in many topological errors during inference. If, however, the network is also tasked to predict the local statistics of the objects surrounding the membrane, focusing merely on the center voxels is no longer sufficient. Instead, the network will have to make use of its entire field of view to predict those statistics. We hypothesize that this leads to more robust internal representations of objects, allowing the network to infer membrane presence from context, even if the local evidence is weak or missing. Many local object statistics are conceivable that would incentivize the network to use its entire field of view. Here, we focus on simple statistics that are efficient to compute during training.

More formally, let $\Omega \subset \mathbb{N}^3$ be the set of voxels in a volume and $y: \Omega \mapsto \{0,...,l\}$ a ground-truth segmentation. A segmentation induces ground-truth affinity values $\text{aff}_N^y$, defined on a voxel-centered neighborhood $N \subset \mathbb{Z}^3$, that is:

$$\text{aff}_N^y : \Omega \mapsto \{0,1\}^{|N|} \quad \text{aff}_N^y(v) = \left(\delta_{y(v)=y(v+n)\neq 0} | n \in N\right) \tag{1}$$

where $\delta$ is the Kronecker function, that is, $\delta_p = 1$ if predicate $p$ is true and 0 otherwise. Our primary learning objective is to infer affinities from raw data $x : \Omega \mapsto \mathbb{R}$, that is, we are interested in learning a function:

$$\text{aff}_N^x : \Omega \mapsto [0,1]^{|N|} \tag{2}$$

such that $\text{aff}_N^x(v) \approx \text{aff}_N^y(v)$.

Similarly to the affinities, we introduce a function to describe the local shape of a segment $i \in \{1, ..., l\}$ under a given voxel $v$. To this end, we intersect the segment $y(v)$ underlying a voxel $v \in \Omega$ with a 3D ball of radius $\sigma$ centered at $v$ to obtain a subset of voxels $S_v \subset \Omega$, formally given as:

$$S_v = \left\{v' \in \Omega \mid y(v) = y(v'), |v - v'|_2^2 \leq \sigma\right\}. \tag{3}$$

We describe the shape of $S_v$ by its size, mean coordinates and the covariance of its coordinates, that is:

$$s(S_v) = |S_v| \tag{4}$$

$$m(S_v) = \frac{1}{s(S_v)} \sum_{v \in S_v} v \tag{5}$$

$$c(S_v) = \frac{1}{s(S_v)} \sum_{v \in S_v} (v - m(S_v))(v - m(S_v))^{\mathsf{T}}. \tag{6}$$

The LSD $\text{lsd}^y : \Omega \mapsto \mathbb{R}^{10}$ for a voxel $v$ is a concatenation of the size, center offset and coordinate covariance, that is:

$$\text{lsd}^y(v) = \left(\underbrace{s(S_v)}_{\text{size}}, \underbrace{m(S_v) - v}_{\text{center offset}}, \underbrace{c(S_v)}_{\text{covariance}}\right). \tag{7}$$

We use $\text{lsd}^y(v)$ to formulate an auxiliary learning task that complements the prediction of affinities. For that, we use the same neural network to simultaneously learn the functions $\text{aff}^x: \Omega \mapsto [0,1]^{|N|}$ and $\text{lsd}^x : \Omega \mapsto \mathbb{R}^{10}$ directly from raw data x, sharing all but the last convolutional layer of the network.

For efficient computation of the target LSDs during training, the statistics above can be implemented as convolution operations with a kernel representing the 3D ball: Let $b^i: \Omega \mapsto \{0, 1\}$ with $b^i(v) = \delta_{y(v)=i}$ be the binary mask for segment $i$ and $w : \mathbb{Z}^3 \mapsto \mathbb{R}$ a kernel acting as a local window (for example, a binary representation of a ball centered at the origin, $w(z) = \delta_{|z|_2^2 \leq \sigma}$). The aggregation of this mask over the window yields the local size $s^i(v)$ of segment $i$ at position $v$. Formally, this operation is equal to a convolution of the binary mask with the local window:

$$s^i(v) = \sum_{v' \in \Omega} b^i(v') w(v - v') = (b^i \times w)(v). \tag{8}$$

To capture the mean and covariance of coordinates as defined above, we further introduce the following voxel-wise functions m and c. Those functions aggregate the pixel coordinates $v$ over the local window w to compute the local center of mass $m^i(v)$ and the local covariance of voxel coordinates $c^i(v)$ for a given segment $i$:

$$
\begin{aligned}
m_k^i(v) &= \frac{(v_k b^i \times w)(v)}{(b^i \times w)(v)} & k \in \{x,y,z\} \\
c_{kl}^i(v) &= \frac{(v_k v_l b^i \times w)(v)}{(b^i \times w)(v)} - m_k^i(v) m_l^i(v) & k,l \in \{x,y,z\}
\end{aligned}
\tag{9}
$$

To obtain a dense volume of shape descriptors, we compute the above statistics for each voxel with respect to the segment this voxel belongs to. Formally, we evaluate for each voxel $v$:

$$\tilde{s}(v) = s^{y(v)}(v) \tag{10}$$

$$\tilde{m}(v) = \left(m_x^{y(v)}(v), m_y^{y(v)}(v), m_z^{y(v)}(v)\right) \tag{11}$$

$$\tilde{c}(v) = \left(c_{xx}^{y(v)}(v), c_{yy}^{y(v)}(v), ..., c_{xz}^{y(v)}(v), c_{yz}^{y(v)}(v)\right) \tag{12}$$

to obtain an equivalent formulation:

$$\text{lsd}^y(v) = (\tilde{s}(v), \tilde{m}(v) - v, \tilde{c}(v)). \tag{13}$$

### Network architectures

We implement the LSDs using three network architectures. The first is a multitask approach, MTLSD, in which the LSDs are output from a 3D U-NET[38], along with nearest neighbor affinities in a single pass. The other two methods, ACLSD and ACRLSD, are both auto-context setups in which the LSDs from one U-NET are fed into a second U-NET to produce the affinities. The former relies solely on the LSDs while the latter also sees the raw data in the second pass (Fig. 1e). We trained networks using gunpowder (http://funkey.science/gunpowder) and TensorFlow (https://www.tensorflow.org/), using the same 3D U-NET architecture[39].

### Post-processing

For affinity-based methods, prediction and post-processing (that is, watershed and agglomeration) used the method described in the previous work[39]. We first passed raw EM data through the networks to obtain affinities. We then thresholded the predicted affinities to generate a binary mask. We computed a distance transform on the binary mask and identified a local maxima. We used the maxima as seeds for

a watershed algorithm to generate an oversegmentation (resulting in supervoxels). We stored each supervoxel center of mass as a node with coordinates in a region adjacency graph (RAG). All nodes of touching supervoxels were connected by edges, which were added to the RAG. In a subsequent agglomeration step, edges were hierarchically merged using the underlying predicted affinities as weights, in order of decreasing affinity, until a given threshold (obtained through a line search on validation data). We extended this method to run in parallel using daisy (https://github.com/funkelab/daisy).

For the FFN network, it was not possible to conduct a standardized comparison owing to the computational power and expertise required to implement the method on the evaluated datasets. Since we were provided with a single segmentation (per dataset)[40], it is not clear what dataset-specific optimizations were done, given that these were production-level segmentations. It is also likely that newer and better FFN segmentations now exist that were not available to compare against at the time we conducted the experiments.

### Reporting summary

Further information on research design is available in the Nature Portfolio Reporting Summary linked to this article.

### Data availability

All datasets analyzed and/or generated during this study (raw data, training data and segmentations) are publicly available (see the 'Data download' notebook on https://github.com/funkelab/lsd). Source data are provided with this paper.

### Code availability

The code used to train networks and segment neurons is available in the 'LSD' repository, https://github.com/funkelab/lsd. Code used to evaluate the results is available in the 'funlib.evaluate' repository, https://github.com/funkelab/funlib.evaluate. All code is free for use under the MIT license.

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

### Acknowledgements

We thank C. Malin-Mayor, W. Patton and J. Buhmann for code contributions; N. Eckstein and J. Buhmann for helpful discussions; S. Berg for code to help with data acquisition; J. Maitin-Shepard for helpful feedback on Neuroglancer; V. Jain, M. Januszewski, J. Kornfeld and S. Plaza for access to data used for training and evaluation. This work was supported by the Howard Hughes Medical Institute. U.M. and A.S. are supported by the Waitt Foundation, Core Grant application NCI CCSG (CA014195), NIH (R21 DC018237), NSF NeuroNex Award (2014862) and the Chan-Zuckerberg Initiative Imaging Scientist Award. T.N. is supported by the Edward R. and Anne G. Lefler Center.

### Author contributions

Conceptualization: S.C.T., J.F. Funding acquisition: S.S., S.C.T., U.M., J.F. Software: A.S., D.D., T.N., J.F. Data consolidation: A.S., D.D., J.F. Evaluation: A.S., J.F. Data dissemination: A.S., J.F. Visualization: A.S., J.F. Writing (original draft): A.S., J.F. Writing (review and editing): A.S., T.N., D.D., W.-C.A.L., U.M., J.F.

### Competing interests

The authors declare no competing interests.

### Additional information

**Extended data** is available for this paper at https://doi.org/10.1038/s41592-022-01711-z.

**Correspondence and requests for materials** should be addressed to Jan Funke.

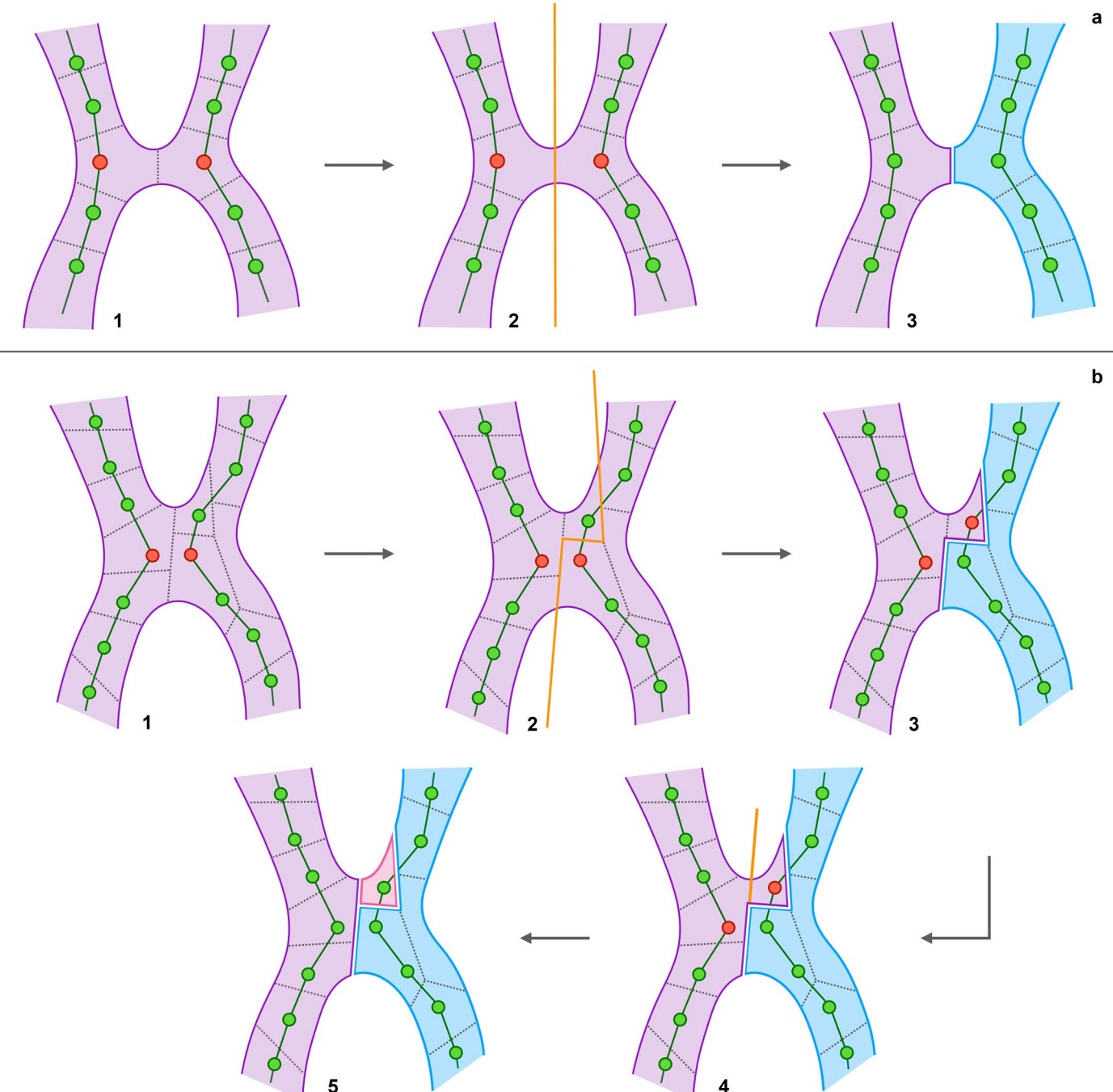

**Extended Data Fig. 1 | Overview of the proposed MCM.** Overview of the proposed MCM. A. Simple case. Two ground-truth skeletons are contained inside an erroneously merged segment. Dashed lines represent supervoxel boundaries and the closest skeleton nodes need to be split to resolve the merge (1). A min-cut is performed (2), resulting in a new segment (3). B. Complex case. Two skeletons are contained in a falsely merged segment as before (1), but the supervoxels are more fragmented. A min-cut is performed (2), resulting in a new segment (3). However, two nodes contained within the original segment need to be split. A second min-cut is performed (4), which produces another segment (5). This results in an additional split error caused by the original cut.

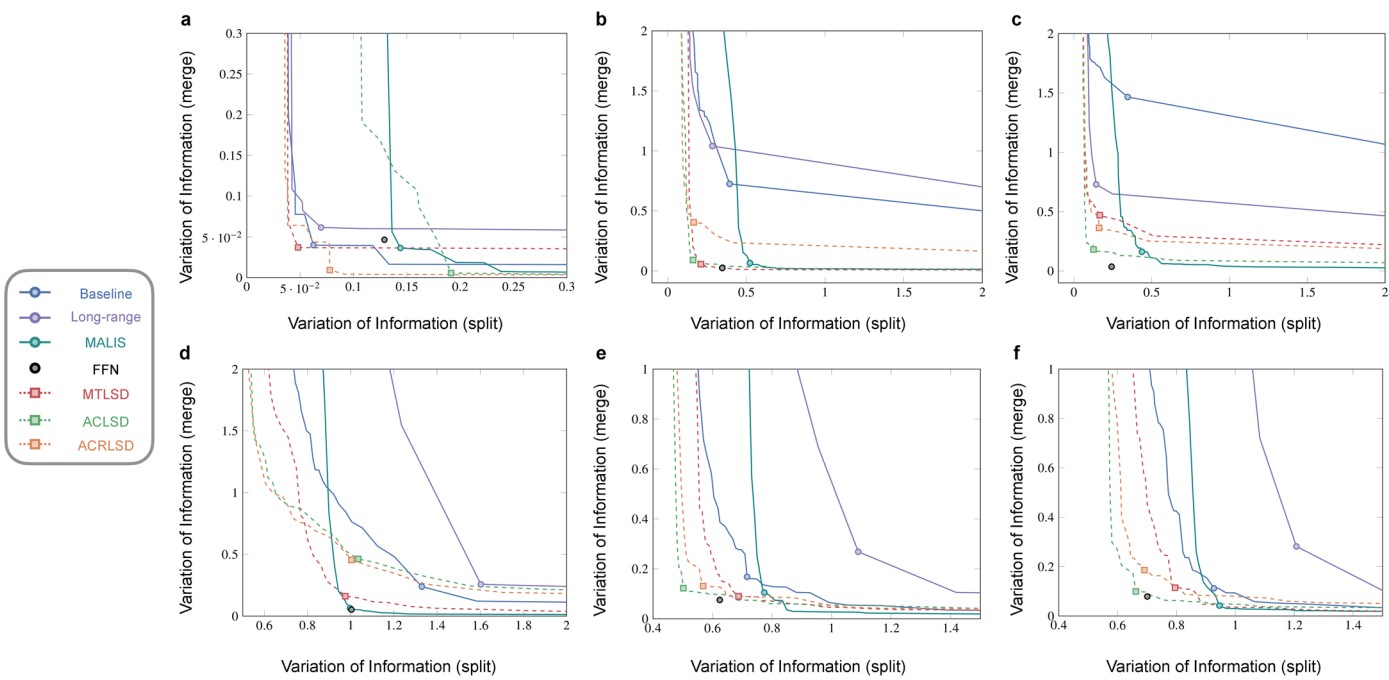

**Extended Data Fig. 2 | Quantitative results on HEMI and FIB-25 datasets.**
Quantitative results on Hemi and FIB-25 datasets. Plot curves show results over range of thresholds. Points correspond to optimal thresholds on testing set, no validation set was available. Lower scores are better. Top row. Hemi dataset. Plot curves show results over range of thresholds for each ROI (A = 12 $\mu$m ROI, B = 17 $\mu$m ROI, C = 22 $\mu$m ROI). Bottom row. FIB-25 dataset. D. Full testing ROI. E,F. Two sub ROIs contained within full ROI.

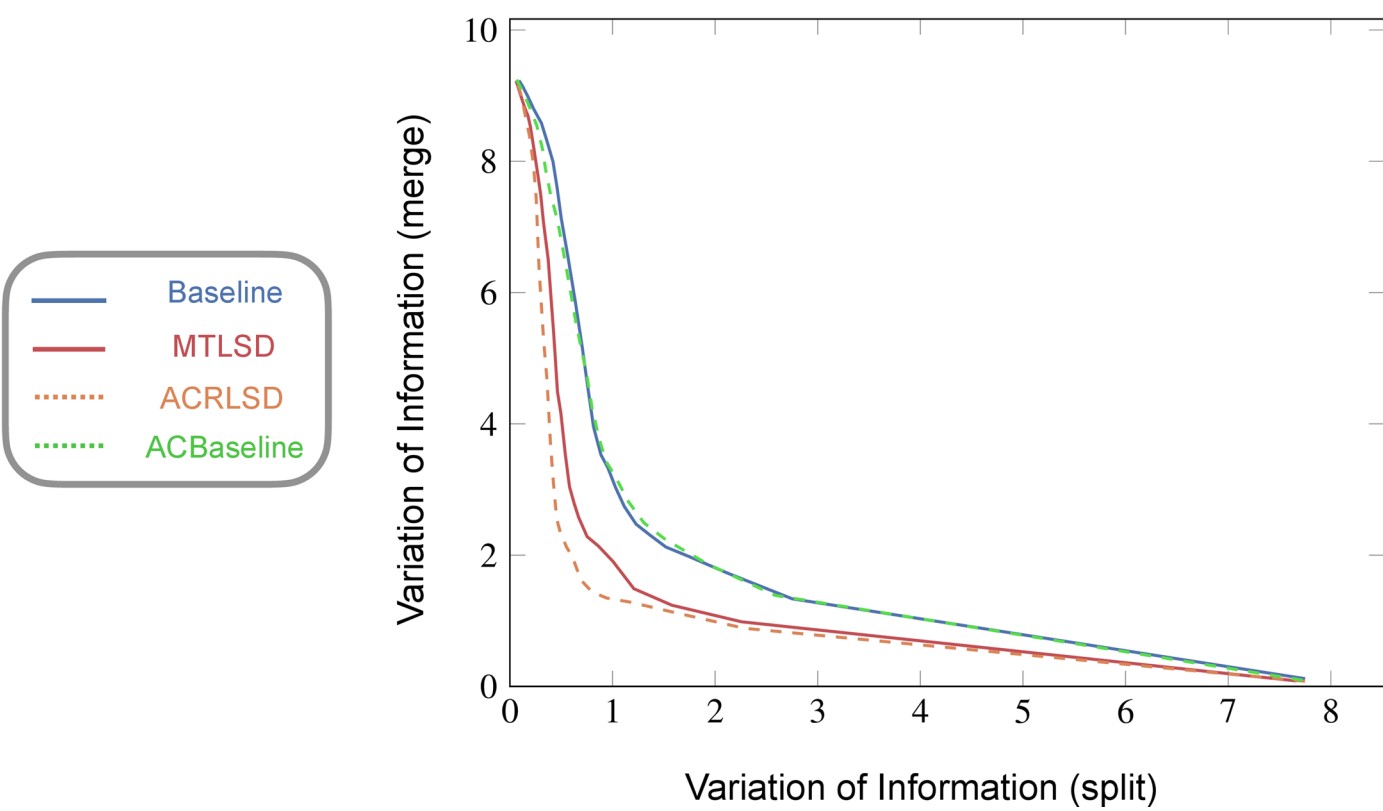

**Extended Data Fig. 3 | Effects of auto-context architecture.** Effects of auto-context architecture. ZEBRAFINCH, benchmark ROI, VoI split versus VoI merge, auto-context comparison.

# Reporting Summary

Nature Research wishes to improve the reproducibility of the work that we publish. This form provides structure for consistency and transparency in reporting. For further information on Nature Research policies, see our Editorial Policies and the Editorial Policy Checklist.

## Statistics

For all statistical analyses, confirm that the following items are present in the figure legend, table legend, main text, or Methods section.

| n/a | Confirmed | |
|---|---|---|
| ☐ | ☒ | The exact sample size (*n*) for each experimental group/condition, given as a discrete number and unit of measurement |
| ☒ | ☐ | A statement on whether measurements were taken from distinct samples or whether the same sample was measured repeatedly |
| ☒ | ☐ | The statistical test(s) used AND whether they are one- or two-sided<br>*Only common tests should be described solely by name; describe more complex techniques in the Methods section.* |
| ☒ | ☐ | A description of all covariates tested |
| ☒ | ☐ | A description of any assumptions or corrections, such as tests of normality and adjustment for multiple comparisons |
| ☒ | ☐ | A full description of the statistical parameters including central tendency (e.g. means) or other basic estimates (e.g. regression coefficient) AND variation (e.g. standard deviation) or associated estimates of uncertainty (e.g. confidence intervals) |
| ☒ | ☐ | For null hypothesis testing, the test statistic (e.g. *F*, *t*, *r*) with confidence intervals, effect sizes, degrees of freedom and *P* value noted<br>*Give P values as exact values whenever suitable.* |
| ☒ | ☐ | For Bayesian analysis, information on the choice of priors and Markov chain Monte Carlo settings |
| ☒ | ☐ | For hierarchical and complex designs, identification of the appropriate level for tests and full reporting of outcomes |
| ☒ | ☐ | Estimates of effect sizes (e.g. Cohen's *d*, Pearson's *r*), indicating how they were calculated |

*Our web collection on statistics for biologists contains articles on many of the points above.*

## Software and code

Policy information about availability of computer code

| Data collection | Standard python packages (networkx, pymongo, scipy, pandas, tensorflow, zarr, gunpowder, daisy), see https://github.com/funkelab/lsd for a full list of requirements, code and tutorials. |
|---|---|
| Data analysis | Standard python packages (pymongo, scipy, pandas, zarr). Custom code, see repositories https://github.com/funkelab/lsd and https://github.com/funkelab/funlib.evaluate for source code, dependencies, installation instructions, and documentation. |

For manuscripts utilizing custom algorithms or software that are central to the research but not yet described in published literature, software must be made available to editors and reviewers. We strongly encourage code deposition in a community repository (e.g. GitHub). See the Nature Research guidelines for submitting code & software for further information.

## Data

Policy information about availability of data

All manuscripts must include a data availability statement. This statement should provide the following information, where applicable:
- Accession codes, unique identifiers, or web links for publicly available datasets
- A list of figures that have associated raw data
- A description of any restrictions on data availability

All used datasets have been previously published, the manuscript contains the corresponding references. In addition, all data generated during this study is available in S3 buckets, see https://github.com/funkelab/lsd, "Data Download" section for details. This section also contains instruction for how to access the raw data.

# Field-specific reporting

Please select the one below that is the best fit for your research. If you are not sure, read the appropriate sections before making your selection.

☒ Life sciences ☐ Behavioural & social sciences ☐ Ecological, evolutionary & environmental sciences

For a reference copy of the document with all sections, see nature.com/documents/nr-reporting-summary-flat.pdf

# Life sciences study design

All studies must disclose on these points even when the disclosure is negative.

| | |
|---|---|
| Sample size | The sample size is given by the number of manually proof-read neuron segmentations (or skeletons) in the available datasets we used for training and evaluation. |
| Data exclusions | No data (from the available datasets) was excluded for analysis. |
| Replication | All source code and data to reproduce the results are or will be publicly available. We did not attempt to replicate the findings, since this is a major computational effort. |
| Randomization | N/A |
| Blinding | N/A |

# Reporting for specific materials, systems and methods

We require information from authors about some types of materials, experimental systems and methods used in many studies. Here, indicate whether each material, system or method listed is relevant to your study. If you are not sure if a list item applies to your research, read the appropriate section before selecting a response.

## Materials & experimental systems

| n/a | Involved in the study |
|---|---|
| ☒ | Antibodies |
| ☒ | Eukaryotic cell lines |
| ☒ | Palaeontology and archaeology |
| ☒ | Animals and other organisms |
| ☒ | Human research participants |
| ☒ | Clinical data |
| ☒ | Dual use research of concern |

## Methods

| n/a | Involved in the study |
|---|---|
| ☒ | ChIP-seq |
| ☒ | Flow cytometry |
| ☒ | MRI-based neuroimaging |

