## [Peer Review File · Nature Methods]

Peer Review Information

Manuscript Title: Local Shape Descriptors for Neuron Segmentation

Corresponding author name(s): Jan Funke

Editorial Notes:

Redactions – unpublished data Parts of this Peer Review File have been redacted as indicated to maintain the confidentiality of unpublished data.

Reviewer Comments & Decisions:

Decision Letter, initial version:

Dear Dr Funke,

Let me first sincerely apologize for the highly unusual delays and my failure to keep you updated. I am very sorry and will strive to avoid any further delays.

Your Brief Communication, "Local Shape Descriptors for Neuron Segmentation", has now been seen by four reviewers. As you will see from their comments below, although the reviewers find your work of considerable potential interest, they have raised a number of concerns. We are interested in the possibility of publishing your paper in Nature Methods, but would like to consider your response to these concerns before we reach a final decision on publication.

We therefore invite you to revise your manuscript to address these concerns. Importantly, the reviewers wonder whether your approach can generalize to serial section EM data and to mouse EM data with their different properties. Please also address the other concerns raised by the reviewers. I'd be happy to discuss a revision plan if you anticipate any issues.

If you need additional space to address the reviewers' concerns, it would be fine to expand the manuscript from a Brief Communication to an Article.

[Redacted] This URL links to your confidential home page and associated information about manuscripts you may have submitted, or that you are reviewing for us. If you wish to forward this email to co-authors, please delete the link to your homepage.

While it is not necessary to adhere to a specific format at this point, I nevertheless wanted to mention a few issues that require your attention at some point.

- Please avoid "Person X et al. show..." or "Fig. Y shows..."
- The introduction and discussion should not have subheadings.
- References should be numbered.
- We cannot accommodate footnotes.
- Please avoid colored text outside of the figures.
- Figures should not contain tables. Please convert these tables into Tables or Supplementary Tables.
- The discussion should not introduce new data or repeat the results.
- When claiming significance, please provide statistical information such as P value and name of statistical test.
- Please convert the Appendix into a Methods section and/or Supplementary Notes.
- The Methods section should follow the Discussion and not be inserted between Introduction and Results.

We hope to receive your revised paper within 4-6 weeks. If you cannot send it within this time, please let us know. In this event, we will still be happy to reconsider your paper at a later date so long as nothing similar has been accepted for publication at Nature Methods or published elsewhere.

OPEN SCIENCE REQUIREMENTS

REPORTING SUMMARY AND EDITORIAL POLICY CHECKLISTS

Please note that these forms are dynamic ‘smart pdfs’ and must therefore be downloaded and completed in Adobe Reader. We will then flatten them for ease of use by the reviewers. If you would like to reference the guidance text as you complete the template, please access these flattened versions at <http://www.nature.com/authors/policies/availability.html>.

DATA AVAILABILITY

We strongly encourage you to deposit all new data associated with the paper in a persistent repository where they can be freely and enduringly accessed. We recommend submitting the data to discipline-specific and community-recognized repositories; a list of repositories is provided here:

<http://www.nature.com/sdata/policies/repositories>

All novel DNA and RNA sequencing data, protein sequences, genetic polymorphisms, linked genotype and phenotype data, gene expression data, macromolecular structures, and proteomics data must be

deposited in a publicly accessible database, and accession codes and associated hyperlinks must be provided in the “Data Availability” section.

Please include a “Data availability” subsection in the Online Methods. This section should inform readers about the availability of the data used to support the conclusions of your study, including accession codes to public repositories, references to source data that may be published alongside the paper, unique identifiers such as URLs to data repository entries, or data set DOIs, and any other statement about data availability. At a minimum, you should include the following statement: “The data that support the findings of this study are available from the corresponding author upon request”, describing which data is available upon request and mentioning any restrictions on availability. If DOIs are provided, please include these in the Reference list (authors, title, publisher (repository name), identifier, year). For more guidance on how to write this section please see: <http://www.nature.com/authors/policies/data/data-availability-statements-data-citations.pdf>

CODE AVAILABILITY

Please include a “Code Availability” subsection in the Online Methods which details how your custom code is made available. Only in rare cases (where code is not central to the main conclusions of the paper) is the statement “available upon request” allowed (and reasons should be specified).

MATERIALS AVAILABILITY

ORCID

Please do not hesitate to contact me if you have any questions or would like to discuss these revisions further. We look forward to seeing the revised manuscript and thank you for the opportunity to consider your work. I would also like to renew my apologies about the delays.

Best regards,
Nina

Nina Vogt, PhD
Senior Editor
Nature Methods

Reviewers' Comments:

Reviewer #1:

Remarks to the Author:

The manuscript "Local Shape Descriptors for Neuron Segmentation" by Sheridan et al. describes an approach to 3D-EM image analysis for connectomics in which locally accessible geometric features of neurites are learned and used for improved axon and dendrite segmentation.

The key value of this approach is the achievement of state-of-the-art reconstruction performance using a manifold computationally more efficient method than the current state-of-the-art (Flood-Filling Networks).

The importance of this improvement in computational efficiency cannot be overstated. Connectomic EM datasets nowadays routinely challenge the ability to store and compute on standard research-level IT infrastructure. The computational load is becoming or has already become the key limitation for the ability to reconstruct connectomes of large brain specimens. Therefore, research to not only improve accuracy but substantially improve efficiency of computational approaches is highly relevant. This is in particular relevant because the development of the so far state-of-the-art approach at the huge commercial company Google - while providing good segmentations - creates a unique market power that is endangering the development of better and more efficient methods in smaller labs. Therefore the efforts of this group should be highly commended to work on these problems in spite of this overshadowing competitor; and the fact that similar results can be reached with orders of magnitude lower computational requirements is a major achievement.

I would have a few remarks that in my view should be considered and improved on for the manuscript to be acceptable at Nature Methods.

1. While the documentation of the computational load improvement is convincing, the detailed quantification of segmentation performance has some remaining questions. In Fig. 3d the best-performing LSD gets close to but does not match the performance of FFNs. In particular this difference is widening again for larger ROIs. It is again commendable that the authors titrated the ERL, the expected run-length-measure, over volume size, because this is one key deficiency of this metric: ERL is dependent on the volume in which it is analyzed! But the notable differences in these two curves should be better explained. It should also be discussed whether this can be called equal performance yet.
2. The shape descriptor approach is being evaluated on three datasets currently, one from the zebrafish brain and two from the drosophila brain. While the focus on the drosophila brain is

understandable it would be extremely helpful to also apply this approach to a key goal of connectomic analyses, which is the mouse brain. In particular the mouse cortex would be a quite important target. This is particularly important since all datasets employed here have a resolution that is quite high (at least in-plane). It would be important to calibrate the performance of LSD on typical large-scale mouse datasets being acquired in the field, as well. In particular, one could recommend to look at the recent dataset from the MICRONS initiative and/or the published data from Motta et al. 2019, Science, from mouse cortex.

3. The introduction focuses heavily on drosophila and songbird results; the mouse brain is underrepresented. This is not necessary and it would contribute to a generalization of the approach to include the literature from other connectomic targets. In particular, this becomes notable when current state-of-the art reconstruction efficacy is reported in the introduction, page 1, right column, last paragraph: it is stated that Lee et al. 2019, this is the Google reconstruction of ssTEM drosophila brain data, has increased human annotation efficacy by a factor of 5.4 to 11.6 minutes per micrometer pathlength. This is quite a notable number – 11.6 minutes per micrometer pathlength if this reviewer is not mistaken would correspond to 11600 minutes per millimeter pathlength, which is close to 200 hours per mm, a quite dramatic investment. In comparison, reconstructions in mouse cortex have been carried out at much faster pace (1-2h/mm, e.g. Boergens et al. 2017 NatMethods; Motta et al., 2019 Science, “Microns” results on biorxive; also the numbers for skeleton-based reconstruction in the fly larval brain are more favorable, see Cardona lab publications). Another example of quite narrow focus is the discussion of “labor required for proofreading” as the major metric for optimization, in which the literature on this topic from the early 2010s is omitted in favor of the authors’ comments in 2016 and 2017. I would suggest a broader view, so to make the manuscript suitable for a non-specialist journal as Nature Methods.

4. The presentation of the figures is quite idiosyncratic (I assume Nature Methods will comment on figure formatting and number of figures for a Brief Communication). But also the results presentation in the figures deserves more care for readability and clarity beyond expert audiences. I suggest avoiding abbreviations wherever possible, at the main axes at least. At the very least abbreviations should be explained in the figure legends (see Fig. 3,4 as examples. VOI ROI ERL etc – enough space at axes to provide human readable labels). Similarly, the usage of nm instead of real-world scales of μm or mm could be avoided.

Reviewer #2:

Remarks to the Author:

The authors propose to include the prediction of segment statistics as auxiliary tasks to the primary training objective.

In particular, the proposed auxiliary objectives include a 10-dimensional vector specifying the size, center coordinates, and correlations of coordinates of each segment within a local window. Compared

to the vanilla U-Nets and U-Nets with extended neighborhood auxiliary tasks (LR), the proposed methods provide an additional performance boost in terms of the segmentation quality. However, there are several major concerns.

1. The idea of utilizing the shape of segments as auxiliary supervision is interesting. However, the technical contribution of the proposed methods is limited. The proposed methods are incremental compared to the closest baseline methods. Adding additional auxiliary tasks is a common technique and use of it in image segmentation has very limited novelty.

2. The readability of Section 2.1 can be improved:

- The equations and formulations in Sec 2.1 are not definitions of the proposed statistics but how the properties are computed from the ground-truth segmentation during implementation. The proposed idea is simple. However, describing the idea with those equations and formulations increases the efforts needed to follow them. I recommend the authors start with definitions (instead of the computation) and descriptions of the three proposed statistics. E.g., $\tilde{s}(v)$ denotes the number of voxels belonging to the same segment of v within a local window.

- All notations in Section 2.1 should be fully defined or made clear, such as Ω (is v the set or Ω the set?) and ℓ . Otherwise, readers have to infer their meaning from context. Besides, the exact shape of subsets such as $N \subset Z^3$ should be made more clear in the method part. Is it a cube, sphere, or other shape?

- Some formulation are not formal and may lead to confusions. For example, for the Kronecker function in Eq 1, do the authors mean $\delta_{\{y(v), y(v+n)\}} \cdot (1 - \delta_{\{y(n+v), 0\}})$ by $\delta_{\{y(v)=y(v+n)\}} \neq 0$?

3. The authors propose to include three statistics of segments as additional supervision. In particular, the size of a segment, the coordinates of the segment center (center of mass), and the covariance of voxel coordinates of the segment. As the motivation for including the three specific properties is not clear, it would be better to include discussions about why each of the three tasks can help the learning of the primary objective.

4. Following 2, the contribution of individual properties (size, center of mass, and correlations) to the final performance are not evaluated and hence their effectiveness is not clear. The authors are recommended to include more ablation studies to analyze their individual contribution to the segmentation task.

5. Too few baseline methods are included in the experiment. Except for the vanilla U-Net and LR as closer baselines, the authors only include FFN in their comparisons. To make the experiment more convincing, I suggest the authors include more recent methods as baselines. [1] provides a good list of related methods.

6. The experimental results indicate that the proposed methods still overall underperforms FFN, although having higher throughput. This limits the practical significance of the proposed methods in computing neuron segmentation, where the segmentation quality is of higher priority.

7. In addition, the performance of adopting the three schemes of LSD (i.e, MTLSD, ACLSD, and ACRLSD) are inconsistent on different datasets and for different evaluation metrics. It makes the proposed methods difficult to generalize to new datasets (or even other tasks). It would be better to provide further discussions and guidance on the choice of LSD schemes given a new dataset to achieve the best performance.

[1] <https://github.com/subeeshvasu/Awesome-Neuron-Segmentation-in-EM-Images>

Reviewer #3:

Remarks to the Author:

The article by Sheridan et. al. investigates a critical and currently unresolved problem in the creation of ‘wiring diagrams’ from large volume EM datasets of brains – connectomics. Specifically, the authors address the issue of segmentation, essentially the algorithmic tracing of neuronal processes, e.g. axons, dendrites, dendritic spines, etc. in those datasets. This is currently the rate limiting step in connectomics and it is critical to discover solutions for the success of connectomics. Specifically, the authors leverage new algorithms based on local shape descriptors (LSD) as a way to segment neurons. Overall, I found the manuscript well written, novel, and likely important for the field.

The main issue I see with the manuscript is that there is very little focus on EM datasets that require alignment. The data analyzed here are exclusively from block face approaches, where data is inherently aligned. This seems a glaring gap since a rationale for LSD approach is that it could scale to petascale volumes yet the 2 current petascale EM volumes do not come from block face approaches but from sectioning individual ultra-thin EM sections and scanned with TEM or SEM, e.g. the IARPA and Harvard datasets as referenced in the manuscript. These approaches require alignment and alignment *prima facie* has profound effects on 3D segmentation. There is little to no description in the manuscript about how alignment could affect LSD based segmentation and this is particularly important in the authors comparison to FFN, one of the main algorithmic ‘competitors, that has been shown to work on previously aligned datasets. This is of obvious importance to much of the connectomics field, both working on petascale and smaller volumes given the prevalence of sectioning approaches.

A second, albeit smaller, issue is that while the MCM metric the authors introduce is interesting and likely important, there is no comparison with FFN using that metric. In addition, the authors make excellent points regarding masking of neuropil, vasculature, etc. increasing the accuracy of segmentation but I could not tell whether the comparisons to FFN on any of the datasets included such masking on the FFN analyses.

Reviewer #4:

Remarks to the Author:

Overall:

“Local Shape Descriptors for Neuron Segmentation” describes a novel loss function for training convolutional neural networks applied to the segmentation of large scale electron microscopy datasets. These data are increasingly important part of cell biology and neuroscience, particularly in the fly and worm communities, and growing influence in the mammalian brain. The principle bottleneck in the field is presently on correcting errors in automated segmentation methods and so novel methods which reduce this cost are important and of broad interest to this field. More broadly, image segmentation is a key step in many biological imaging applications so ideas about how to improve it are of broad interest. The authors do in my opinion a thorough set of computational experiments to compare the performance of their method to other previously presented methods, holding most of the methods constant. They have done a good job packaging and presenting their code and data and made it all transparently available. The weaknesses I found in the paper are in that they haven’t compared their results to existing segmentation results, which leaves open the possibility that authors of alternate methods have optimized various hyper-parameters to synergize with their work. Furthermore, they only analyze FIB data, and since many of the papers cited in the introduction use serial section data with higher SNR but lower axial resolution it will leave readers wondering if that is important or not in selecting a method for their dataset. Finally, I think the computational cost argument is a compelling one, and think the authors should go further in making it more explicit in estimating the integrated costs of a computational and proofreading pipeline that uses these methods. I have a number of relatively minor questions and suggestions. I think it likely that with revisions I would strongly recommend this paper for publication.

Major issues:

The authors do an admirable job of trying to do a systematic evaluation of different methods that are present in the literature across a range of datasets. This is however a very complex landscape and I think there are some ways that some checks should be done.

For example, in each of the proposed methods they have implemented their own versions of the networks with their own training regimes. This is the optimal approach to compare the specific computational method being proposed here, while holding all other things constant (such as hyperparameters of the UNET, augmentation procedures, agglomeration threshold, any post-processing or heuristics steps on agglomeration, or contextual masking procedures). However, it's not clear whether there are interactions between these things and the methods chosen. Computational researchers are doing their own meta-optimizations and might not even be aware of the interactions between their methods. For that reason, I think it would also be useful to compare their results on datasets in which publicly available segmentations of the data are also available, trained and optimized by the authors of the method that is being compared to. If there are differences in the performance that aren't in the more controlled computational experiments shown here, it is useful to understand that either factors outside of the loss function proposed or interactions between the loss function and these other procedures are driving meaningful differences in performance. If this is difficult to obtain, I think at least discussing these range of issues in the context of the great work the authors have done to release the precise methods used for training and running inference on their networks.

Second, one issue is that the datasets compared to are isolated to FIB methods with more isotropic resolution, as opposed to serial section methods with generally higher SNR but lower axial resolution, whereas several of the methods they are comparing to were designed for serial section methods. This is mentioned very briefly in one sentence in the discussion. Whether there is an interaction between methods and data modality is I think an open question in the field and should either be addressed by comparing the methods on some other acquisition types. If this method is going to be of wide utility to the field, it would seem testing on at least one serial section dataset should be done.

Third, and this is maybe related to the above point. I was struck how there was little mention of how LSDs might interact with alignment errors. I think there are a few reasonable choices to be made in how you would do data augmentation that introduced misalignments, and there could be an argument for using 2d LSDs instead of or in addition if it ends up being easier to predict them in the face of alignment errors. Perhaps in the FIB data this is less of an issue, but since this proposes to be a general method I think it's worth discussing.

Given that the argument for this method is one of computation efficiency, it seems to me that the authors have gathered the data to map out the space of economics about where these methods lie. For example, given a cost per teraflop, and a cost per edit, the total cost of each computational pipeline can be evaluated. Although the precise values of those values might be ambiguous, there is data now on how many edits were applied by roughly how many human hours of effort on some large scale dataset. Using a reasonable ranged approach of the relative costs of those two things, one could compare the landscapes of costs across the methods.

To sketch out a very simple version of this analysis. If humans perform ~ 50 edits per hour and cost \$15-50 dollars an hour, then edits cost \$.3-\$1. For the largest of the small volumes that puts the edit cost in the \$60-1000 range. The cost of FFNs for those is $1.6 \cdot 10^4 \text{ um}^3 / .07 \text{ um}^4 \text{ per second} = 228,571$ seconds or 63 hours. If a nvidia v100 is rentable on google cloud for example at \$0.74-2.48 per hour, the total cost is \$46-150 dollars, making the computational costs within the same order of magnitude of the proofreading costs. I think the authors are positioned to make a much more systematic and careful estimate than I've done here, but I think putting these costs into such an explicit framework to be traded off would move the thinking of the field forward.

Minor issues:

Why is there not a FFN line in Figure 3B?

Shouldn't you plot the ERL of the ground truth data after cropping to put the fluctuations and trends observed in the experimental segmentations into context? The discussion seems to suggest that cropping is a large driver of the observed fluctuations but I'm not sure that is true, seems plotting it would make it clear one way or the other.

I think the section on merge errors is correct, but there is an added problem introduced in the agglomeration step where no agglomeration algorithm exists that doesn't merge objects at all contact sites even when the merge error only exists at one site. I think this could be explained more clearly in the text, likely after the concept of agglomeration is introduced.

It is not clear to me that the shape descriptors chosen reflect the entire space of potentially useful accessory learning tasks, nor whether including all of them is important to performance. The authors might provide some perspective in discussion on these points, and observations on how the number of LSD features included affects the throughput of the network. The central argument of the methods utility is that it is more efficient, there is little relatively little work described on attempting to optimize the efficiency of the network.

Since the submission of the paper there are two mm^3 preprints (Microns Consortium et al, and Shapson-Coe et. al.) that might be better citations for the large dataset section. There is a Turner et al citation for synapse detection that should be included (<https://arxiv.org/abs/1904.09947>).

The min-cut metric is a good one and its more than hypothetical in the sense that min-cuts on the agglomeration graph are the mechanism behind the splitting tool used in the PyChunkedGraph, which seems worth mentioning.

This is more speculation than a point that must be addressed... have you tried to use a purely topological definition of split/merge operations on skeletons to define a min-cut. If you used a synapse aware skeletonization technique you could define it in terms of how many cuts are required on the skeleton to make the synapses appear in exactly the right components. Maybe that is more tractable than the agglomeration version and could be applied to FFNs.

Finally, I do agree with the authors that this method likely does have applications outside of EM segmentation. I think the paper could broaden its interest if they invested time in applying it to one problem area outside of electron microscopy. There are a number of open source competitions and datasets for segmentation of fluorescence images which could be used for this purpose.

Author Rebuttal to Initial comments

We thank the reviewers and the editor for their helpful comments on our manuscript.

The main points brought up in the reviews concerned the applicability of LSDs beyond the dataset we already evaluated (focused ion beam scanning EM of *Drosophila* and serial block-face EM of zebrafish) as well as a more in-depth study of the LSD components. We have now included results on mouse cortex, imaged using serial section transmission EM (see Supplementary Section F.1). Furthermore, we now also investigate the usefulness of LSDs for non-neural tissue by including a plant epithelial tissue dataset in our analysis (see Supplementary Section F.3). We also included an ablation study to highlight the importance of the chosen LSD components (see Supplementary Section F.2).

The following sections address individual points brought up in the reviews, following the same order. Changes to the manuscript and the Supplemental Material are underlined and contain a pointer to the corresponding section in this document.

Reviewer 1

Accuracy Comparison of LSD and FFN

R1:1 One of the contributions of our manuscript is to demonstrate how two commonly used error metrics (VOI and ERL) relate to proof-reading effort (i.e., the amount of edit operations needed to correct a segmentation). We find that VOI is a considerably more reliable proxy for proof-reading effort than ERL (see Figure 3B, 3D, Figure 18, and Section 4.1). In particular, we observe similarities between VOI and MCM, the latter being a metric to count the number of needed edit operations in contemporary proof-reading tools. In light of those findings regarding VOI as a proxy for proof-reading effort, we find it justified to measure the overall accuracy of either method with VOI.

We have now clarified in our main text that we consider the accuracy of LSD-based methods to be on par with FFN only with respect to VOI. We are not on par with respect to ERL, which is likely because even small merge errors can contribute substantially to ERL (see Figure 18B and 18D). In short, ERL considers a whole neuron to be segmented incorrectly if it was merged with a fragment from another neuron, which can potentially lead to erratic scores that no longer reflect the amount of time needed to proof-read the resulting segmentation.

Those observations also explain the differences between the ERL scores of the LSD-based methods and FFN (e.g., Figure 3D). AcRLsd produces more merges (VOI merge 1.436) than FFN (VOI merge 1.118). Those merge errors contribute disproportionately to the ERL computation and depend on the size of the neurons involved in the merge (e.g., merging two small neuron fragments causes a smaller decrease in ERL than merging two large fragments, although the effort needed to split them is presumably the same). We added a brief discussion of those discrepancies to Section 3.4.1.

Evaluation on Mouse Dataset

R1:2 To demonstrate the usability of LSDs on mouse cortex, we evaluated them on publicly available data from the MICrONS Consortium (Microns Consortium et al., 2021). Results are shown in Supplementary Section G.1 and indicate that LSDs are not limited to *Drosophila* and Zebrafinch tissue. Consistent with results on the other datasets, an AcLsd approach yielded superior accuracy to alternative methods. It is important to note that the available testing data is likely too small to accurately extrapolate performance on much larger datasets (as discussed in Section 4.5). However, it is reasonable to assume that the LSDs would also benefit from the same preprocessing that is commonplace for large sSTEM datasets (such as alignment, contrast adjustment, and masking).

Proof-Reading Times

We revised the introduction to include references to reported tracing times in mouse and larval tissue from previous publications. Due to the challenging nature of dense invertebrate neuropil, which contains many fine-tip neurites, it is reasonable to expect slower tracing times on average than in mammalian tissue.

Figures

We have updated the figures in the main text to avoid abbreviations on the main axes (e.g., ROI-> Region of Interest [μm^3]), and have changed the ERL units from nanometers to microns in the tables.

Reviewer 2

Novelty

We do not consider the novelty of local shape descriptors as an auxiliary learning task to be our main contribution. Our main contribution is the finding that with such a simple method we are able to propel affinity-based methods to be on par with the current state of the art of neuron segmentation, at a substantially decreased computational cost. In that context, we consider the simplicity of the method a

plus, as it allows other labs to easily implement our method for use on their own datasets, using commonly available hardware infrastructure.

We would also like to note that the Multi-Cut Metric (MCM) we developed to assess common error metrics in our field is indeed novel and non-trivial. The MCM measures the number of edit operations a proof-reader would have to perform using contemporary tools that allow to merge and split neuron segmentations with a few clicks.

Clarity of Method Section

We revised the method section to first introduce a clear definition of the used statistics underlying the LSDs, and then discuss the computation using convolutions with a 3D ball. Other minor edits ensure that the used notation is introduced (e.g., the Kronecker delta) and that the exposition is easier to follow. All changes in Section 2.1 are highlighted.

Discussion of Auxiliary Learning Task

The main rationale for our selection of the LSD components is to encourage the neural network to make use of its entire field of view (FOV) to reach a decision about the presence or absence of a boundary in the center of the FOV. The following might provide an intuition for why we thought this to be helpful: In many cases, the intensity value of the voxel in the center of the FOV is a good indicator for the presence of a boundary, as many EM staining protocols target lipid bilayers. Trained naively, a neural network might focus only on a few center voxels to detect membranes and achieve very good accuracy during training, especially if trained using a voxel-wise loss. However, this strategy will fail in cases where boundary evidence is ambiguous. Those cases might be rare from the point of view of training a network (they might constitute less than one percent of samples), but given the large size of datasets in connectomics, those cases still result in many topological errors overall. If, however, the network is also tasked to predict the local statistics of the objects surrounding the membrane, focusing merely on the center voxels is no longer sufficient. The network will have to make use of its entire field of view to predict those statistics. We hypothesize that this leads to more robust internal representations of objects, allowing the network to “see” a membrane based on context, even if the local evidence is weak or missing. This in turn helps to improve the boundary prediction by correlating, e.g., the center of mass vectors with the target affinities.

We added this intuition and a brief motivation for our choice of statistics to Section 2.1.

Ablation Study on LSD Components

R2:4 We conducted an ablation study of LSD components on the smallest RoI from the Hemi-brain dataset (shown in Supplementary Section F.2). We found that the combination of components in an MtLsd network generally is not imperative to performance. This is likely due to combining the LSDs loss with the affinities loss. The second pass of an AcLsd network works well with any combination of components as input, with the exception of the size of the neural process as the sole descriptor. However, it is likely that on much larger datasets, using all components of the LSDs becomes

necessary. These results suggest that in some instances, using a subset of descriptors is sufficient for improving affinities.

Comparison to Other Neuron Segmentation Methods

R2:5 We thank the reviewer for pointing us to this resource (github.com/subeeshvasu/Awesome-Neuron-Segmentation-in-EM-Images), which provides a comprehensive list of approaches to neuron segmentation. We considered each method dating back to at least 2019 and believe that it is not necessary to evaluate against any additional approaches not already included in our manuscript.

First, not all methods listed in the resource are neuron segmentation methods for EM. Some references present post-processing methods, which are complementary to neuron segmentation methods. In our manuscript, we intend to keep the post-processing the same to evaluate neuron segmentation methods under identical conditions. Other references are review papers, apply to light microscopy images, address the problem of image restoration, or proof-reading of segmentation results. In Table 1 in this reply we provide a categorization of all methods dating back to 2019.

Regarding actual neuron segmentation methods for EM, we make a distinction between methods that are ready to be applied to large EM volumes (Januszewski et al., 2018; Funke et al., 2019) and proof-of-concept implementations of (potentially promising) new approaches (Cerrone et al., 2019; Meirovitch et al., 2019; Linsley et al., 2020; Luther and Seung, 2019; Lee et al., 2019a). We already compare against the former in our manuscript. Notably, FFN (Januszewski et al., 2018) constitutes the current state of the art and we believe that a comparison against this method is sufficient for large-scale EM segmentation. Of the proof-of-concept methods, only the recent dense metric learning approaches show the potential to compete in accuracy with the current state of the art (Luther and Seung, 2019; Lee et al., 2019a). We agree that a comparison would be of great interest.

[REDACTED]

Practical Significance

It appears we have been a bit too modest in discussing our results. Although it is true that LSD-based methods underperform compared to FFN in terms of the ERL metric, we match performance with respect to the Vol metric. The discrepancy between those metrics is concerning, which is why we devised the Multi-Cut Metric (MCM) to measure the amount of proof-reading time needed to correct an automatic segmentation. Crucially, the MCM assumes a proof-reading tool in which merge and split errors can be fixed with a few clicks. Those tools are now routinely used for large-scale connectome reconstructions (e.g., NeuTu and FlyWire). By comparing Vol and ERL to MCM, we find that ERL is a poor proxy for proof-reading effort, mainly because ERL disproportionately punishes merge errors.

We have now made this point more clear in our manuscript. Please refer to the response R1:1 of reviewer 1, who raised a similar concern.

Given that LSD-based methods achieve similar accuracy under Vol as FFN, the computational efficiency of LSD-based methods does indeed have a high practical significance. On the one hand, it allows many

labs to perform comparative connectomics on their own compute infrastructure. Since the publication of our preprint, we are already aware of several labs using LSD-based segmentation on their datasets ([REDACTED]). On the other hand, we believe that even large private entities with vast resources will welcome the computational speedup for the processing of future petabyte-scale datasets.

Choice of LSD Scheme

We tested three LSD schemes (MtLsd, AcLsd, AcRLsd) and found variable results on the smaller testing datasets. However, it is clear that an auto-context approach is increasingly useful when scaling to larger datasets. The slight increase in compute is arguably worth the significant increase in accuracy. On smaller datasets, it is probably sufficient to save compute with a multi-task approach. Each scheme seems to consistently outperform baseline methods on all investigated datasets. It still holds that the best scheme per dataset would ideally be chosen based on results on a validation set when available (as was done for the Zebrafinch). We added a comment to address this in Section 4.3.

Category	Methods
Large-Scale Neuron Segmentation Methods	Januszewski et al. (2018) ; Funke et al. (2019)
Proof-of-Concept Methods	Cerrone et al. (2019) ; Meirovitch et al. (2019) ; Linsley et al. (2020) ; Luther and Seung (2019) ; Lee et al. (2019a)
Post-processing	Bailoni et al. (2019) ; Matejek et al. (2019) ; Hascoet et al. (2019) ; Zhuang et al. (2020)
Proofreading tools	Urakubo et al. (2019) ; Hubbard et al. (2020)
Image restoration	Ishii et al. (2020)
Light microscopy	Gornet et al. (2019)
Review papers	Motta et al. (2019) ; Lee et al. (2019b)
Dissertations	Staffler (2020) ; Wolf (2020)

Table 1: Categorization of methods listed on <https://github.com/subeeshvasu/Awesome-Neuron-Segmentation-in-EM-Images>.

Reviewer 3

Evaluation on ssTEM Dataset

Currently, serial section datasets remain the largest volumes to process. However, FIB-SEM is not limited to small volumes (e.g the Hemi-brain brain) and larger datasets will continue to be acquired. While the alignment of ssTEM datasets poses a concern for all segmentation algorithms, it has been shown that combinations of global and local alignment help to improve the accuracy of networks (Dorkenwald et al., 2019; Li et al., 2019; Turner et al., 2020; Microns Consortium et al., 2021). Consequently, as the quality of alignment improves, it becomes less of a challenge for future segmentation algorithms. Additionally,

increased anisotropy in ssTEM data poses a challenge for networks. We found that the LSDs not only out-performed other methods on isotropic data (Fib-25, Hemi-brain), but also on larger anisotropic data (Zebrafinch). However, the anisotropy factor of 2 might not be indicative of performance on higher anisotropy commonly seen in serial section data. To address these concerns, we evaluated the LSDs on publicly available mouse tissue from the MICrONS Consortium (Microns Consortium et al., 2021). Results are shown in Supplementary Section F.1 and indicate that LSDs can also be used for serial section data. We also briefly elaborate on this in response to review point R1.2 above.

Alignment of Datasets

R3:2 Alignment is a necessary preprocessing step for generating production segmentations on serial section data (Dorkenwald et al., 2019; Li et al., 2019; Turner et al., 2020; Microns Consortium et al., 2021). As far as we are aware, all methods benefit from an improved alignment. If using the LSDs to produce a production segmentation on ssTEM data, it is likely that they will also fail if careful consideration is not taken into first properly aligning the data.

MCM Metric on FFN Segmentation

R3:3 Since FFN generates large segmentations (and therefore no fragment graph), it can not trivially be broken down into smaller fragments, it is unfortunately not possible to evaluate the Min- Cut Metric (MCM) on FFN. We address this now at the end of Section 3.4.1.

Masking of FFN Segmentation

R3:4 All segmentations (including FFN) used some form of masking across datasets. On Fib-25, an irregularly shaped neuropil mask was used which comprised the majority of the raw data and excluded most background regions. For the Hemi-brain, a more accurate mask was used that confined segmentations to the Ellipsoid Body neuropil. Most notably, the Zebrafinch dataset used a more complex mask which ignored cell bodies, blood vessels, myelin, and background regions. These masks ensured that evaluated segmentations were restricted to neuropil.

Reviewer 4

Comparison to Existing Segmentations

A comparison of complete connectome reconstruction pipelines on public benchmark data would certainly be useful. In our manuscript, however, we compare different methods for one step of this pipeline, i.e., the impact of the neural network that predicts a neuron segmentation. There are many other variables outside of neural networks that contribute to the accuracy of the whole pipeline (e.g., alignment, masking, and post-processing). Therefore, the amount of engineering put into specific datasets will greatly influence the performance of differing pipelines.

Given these constraints, comparing against publicly available segmentations does not allow us to evaluate pure neuron segmentation accuracy across methods. Since neuron segmentation accuracy is orthogonal to other contributing factors (i.e., a better segmentation will always improve downstream

results), we prefer to compare against existing methods under the same conditions. We nevertheless included FFN, although it uses a different training procedure, since it is the current state of the art in neuron segmentation. However, even in the case of FFN, we made sure to keep as many pipeline components as equal as possible (e.g. raw data, training data, and masks). We added a brief discussion to Section 3.1.

Evaluation on ssTEM Dataset

To address the accuracy of LSDs on serial section data, we evaluated the LSDs on publicly available mouse tissue from the MICrONS Consortium (Microns Consortium et al., 2021). Results are shown in Supplementary Section G.1 and indicate that LSDs can also be used for serial section data. We also briefly elaborate on this in response to review points R1.2 and R3.1 above.

Alignment of Datasets

Alignment and anisotropy of ssTEM datasets is a practical concern for all segmentation algorithms. As with other methods, LSDs will likely perform poorly on non-aligned data. We also briefly discuss this in review point R3.2.

Integrated Costs of Computation and Proof-Reading

Relating the costs of proof-reading to computation of neuron segmentations is an excellent idea. We contemplated how our results could provide insights and realized that we are not in a position to provide a simple answer.

First, the field is still moving quite quickly in terms of proof-reading tools. Tools like NeuTu and FlyWire are very recent and are still being improved (Zhao et al., 2018; Dorkenwald et al., 2022). Additional information like microtubule tracks, unsupervised cell typing, and biological priors will likely be integrated to decrease proof-reading times in the future. Similarly, higher-level machine learning methods are developed to spot and fix reconstruction errors by learning from the connectomics data already available. We think it is reasonable to assume that proof-reading costs will substantially decrease in the near future, rendering any analysis that we could perform right now obsolete.

Second, we found it surprisingly hard to put a single number on the cost of proof-reading of neuron segmentations, even if we assume that current tools do not improve. Proof-reading times vary greatly between different organisms, brain regions, and even cell types; as well as between different imaging modalities. Furthermore, proof-reading usually also includes correcting the automatic detection of synapses, which is unrelated to fixing segmentation errors.

Finally, the amount of time (and therefore money) spent on proof-reading depends highly on the biological question at hand. For many questions (especially in the emerging field of comparative connectomics), it will be sufficient to reconstruct certain circuits of interest, instead of the whole connectome. This could easily cut the proof-reading time down by several orders of magnitude. The cost of running an automated segmentation to reconstruct brain-spanning neurons will, however, not be affected by that.

In conclusion, we think that this topic deserves a much more refined discussion (in the form of an opinion piece, for example), which exceeds the scope of our manuscript.

FFN in Figure 3b

R4:5 We added the FFN plot in for Roi size vs Vol sum on the first three Rols of the Zebrafinch dataset (see Figure 3.B).

ERL of Ground-Truth Data

R4:6 We added this plot to the supplementary (Figure 19) and note it in Section 3.4.1.

Merge at Multiple Sites

R4:7 Commonly used agglomeration algorithms (including the one we use here) do in fact merge only at a single site. The underlying algorithm is referred to as “single-linkage clustering”, and works by growing a minimal spanning tree on the adjacency graph of fragments. For that, edges are scored based on the affinities between the incident fragments.

The reason why it appears that segments are merged on multiple sites is a visualization limitation. In principle, data structures exist that would allow a proof-reading tool to draw a boundary between two locally not merged fragments, even if they have the same label because they are connected elsewhere. We have added a note to Section 3.2.3 to explain that merge errors can be solved with a single interaction, regardless of the number of contact sites.

Impact of Individual LSD Components

R4:8 It is true that the LSDs do not reflect the entire space of potential descriptors. We agree that future work could aim to find an optimal embedding rather than a hand-engineered one, and discuss this point in Section 5. Since the LSDs only add a few feature maps to the output of the U-Net, they only add a negligible computational overhead with respect to Baseline affinities. The majority of features are accumulated in higher layers of the U-Net. We have added a comment to address this in Section 3.5. We have also addressed the rationale of the LSDs and their components in review points R2.3 and R2.4.

Additional References

We have added appropriate additional references for large dataset acquisition in Section 1.

Mention PyChunkedGraph

We have added appropriate references to proofreading tools that use graph cuts in Section 3.2.

Non-EM Applications

To demonstrate usability of LSDs outside the scope of neuron segmentation, we evaluated them on publicly available plant epithelial cell data from Wolny et al. (2020). We discuss the results in Section 5 and Supplementary Section F.3. Our findings suggest that LSDs are applicable to other data modalities,

and are generally agnostic to object shape. This could potentially have implications for challenging datasets in which there are combinations of spherical and elongated objects.

References

- Bailoni, A., Pape, C., Wolf, S., Beier, T., Kreshuk, A., and Hamprecht, F. A. (2019). A Generalized Framework for Agglomerative Clustering of Signed Graphs applied to Instance Segmentation. arXiv:1906.11713 [cs]. arXiv: 1906.11713.
- Cerrone, L., Zeilmann, A., and Hamprecht, F. A. (2019). End-To- End Learned Random Walker for Seeded Image Segmentation. In 2019 IEEE/CVF Conference on Computer Vision and Pattern Recognition (CVPR), pages 12551–12560, Long Beach, CA, USA. IEEE.
- Dorkenwald, S., McKellar, C. E., Macrina, T., Kemnitz, N., Lee, K., Lu, R., Wu, J., Popovych, S., Mitchell, E., Nehoran, B., Jia, Z., Bae, J. A., Mu, S., Ih, D., Castro, M., Ogedengbe, O., Halageri, A., Kuehner, K., Sterling, A. R., Ashwood, Z., Zung, J., Brittain, D., Collman, F., Schneider-Mizell, C., Jordan, C., Silversmith, W., Baker, C., Deutsch, D., Encarnacion-Rivera, L., Kumar, S., Burke, A., Bland, D., Gager, J., Hebditch, J., Koolman, S., Moore, M., Morejohn, S., Silverman, B., Willie, K., Willie, R., Yu, S.-c., Murthy, M., and Seung, H. S. (2022). Flywire: online community for whole-brain connectomics. *Nature Methods*, 19(1):119–128.
- Dorkenwald, S., Turner, N. L., Macrina, T., Lee, K., Lu, R., Wu, J., Bodor, A. L., Bleckert, A. A., Brittain, D., Kemnitz, N., Silversmith, W. M., Ih, D., Zung, J., Zlateski, A., Tartavull, I., Yu, S.-C., Popovych, S., Wong, W., Castro, M., Jordan, C. S., Wilson, A. M., Froudarakis, E., Buchanan, J., Takeno, M., Torres, R., Mahalingam, G., Collman, F., Schneider-Mizell, C., Bumbarger, D. J., Li, Y., Becker, L., Suckow, S., Reimer, J., Tolia, A. S., Costa, N. M. d., Reid, R. C., and Seung, H. S. (2019). Binary and analog variation of synapses between cortical pyramidal neurons. *bioRxiv*, page 2019.12.29.890319. Publisher: Cold Spring Harbor Laboratory Section: New Results.
- Funke, J., Tschopp, F., Grisaitis, W., Sheridan, A., Singh, C., Saalfeld, S., and Turaga, S. C. (2019). Large Scale Image Segmentation with Structured Loss Based Deep Learning for Connectome Reconstruction. *IEEE Transactions on Pattern Analysis and Machine Intelligence*, 41(7):1669–1680.
- Gornet, J., Venkataraju, K. U., Narasimhan, A., Turner, N., Lee, K., Seung, H. S., Osten, P., and Sümbül, U. (2019). Reconstructing neuronal anatomy from whole-brain images. Technical Report arXiv:1903.07027, arXiv. arXiv:1903.07027 [cs, q-bio] type: article.
- Hascoet, T., Metge, B., Takiguchi, T., and Ariki, Y. (2019). Entropy policy for supervoxel agglomeration of neurite segmentation. page 6.
- Hubbard, P. M., Berg, S., Zhao, T., Olbris, D. J., Umayam, L., Maitin-Shepard, J., Januszewski, M., Katz, W. T., Neace, E. R., and Plaza, S. M. (2020). Accelerated EM Connectome Reconstruction using 3D Visualization and Segmentation Graphs. preprint, Neuroscience.
- Ishii, S., Lee, S., Urakubo, H., Kume, H., and Kasai, H. (2020). Generative and discriminative model-based approaches to microscopic image restoration and segmentation. *Microscopy*, 69(2):79–91.

- Januszewski, M., Kornfeld, J., Li, P. H., Pope, A., Blakely, T., Lindsey, L., Maitin-Shepard, J., Tyka, M., Denk, W., and Jain, V. (2018). High-precision automated reconstruction of neurons with flood-filling networks. *Nature Methods*, 15(8):605.
- Lee, K., Lu, R., Luther, K., and Seung, H. S. (2019a). Learning Dense Voxel Embeddings for 3D Neuron Reconstruction. arXiv:1909.09872 [cs]. arXiv: 1909.09872.
- Lee, K., Turner, N., Macrina, T., Wu, J., Lu, R., and Seung, H. S. (2019b). Convolutional nets for reconstructing neural circuits from brain images acquired by serial section electron microscopy. *Current Opinion in Neurobiology*, 55:188–198.
- Li, P. H., Lindsey, L. F., Januszewski, M., Zheng, Z., Bates, A. S., Taisz, I., Tyka, M., Nichols, M., Li, F., Perlman, E., Maitin-Shepard, J., Blakely, T., Leavitt, L., Jefferis, G. S. X. E., Bock, D., and Jain, V. (2019). Automated Reconstruction of a Serial-Section EM Drosophila Brain with Flood-Filling Networks and Local Realignment. bioRxiv, page 605634. Publisher: Cold Spring Harbor Laboratory Section: New Results.
- Linsley, D., Kim, J., Berson, D., and Serre, T. (2020). Robust neural circuit reconstruction from serial electron microscopy with convolutional recurrent networks. arXiv:1811.11356 [cs]. arXiv: 1811.11356.
- Luther, K. and Seung, H. S. (2019). Learning Metric Graphs for Neuron Segmentation In Electron Microscopy Images. arXiv:1902.00100 [cs].
- Matejek, B., Haehn, D., Zhu, H., Wei, D., Parag, T., and Pfister, H. (2019). Biologically-Constrained Graphs for Global Connectomics Reconstruction. In 2019 IEEE/CVF Conference on Computer Vision and Pattern Recognition (CVPR), pages 2084–2093, Long Beach, CA, USA. IEEE.
- Meirovitch, Y., Mi, L., Saribekyan, H., Matveev, A., Rolnick, D., and Shavit, N. (2019). Cross-Classification Clustering: An Efficient Multi-Object Tracking Technique for 3-D Instance Segmentation in Connectomics. In 2019 IEEE/CVF Conference on Computer Vision and Pattern Recognition (CVPR), pages 8417–8427, Long Beach, CA, USA. IEEE.
- Microns Consortium, ., Bae, J. A., Baptiste, M., Bodor, A. L., Brittain, D., Buchanan, J., Bumbarger, D. J., Castro, M. A., Celii, B., Cobos, E., Collman, F., Costa, N. M. d., Dorkenwald, S., Elabbady, L., Fahey, P. G., Fliss, T., Froudarakis, E., Gager, J., Gamlin, C., Halageri, A., Hebditch, J., Jia, Z., Jordan, C., Kapner, D., Kemnitz, N., Kinn, S., Koolman, S., Kuehner, K., Lee, K., Li, K., Lu, R., Macrina, T., Mahalingam, G., McReynolds, S., Miranda, E., Mitchell, E., Mondal, S. S., Moore, M., Mu, S., Muhammad, T., Nehoran, B., Ogedengbe, O., Papadopoulos, C., Papadopoulos, S., Patel, S., Pitkow, X., Popovych, S., Ramos, A., Reid, R. C., Reimer, J., Schneider-Mizell, C. M., Seung, H. S., Silverman, B., Silversmith, W., Sterling, A., Sinz, F. H., Smith, C. L., Suckow, S., Takeno, M., Tan, Z. H., Tolia, A. S., Torres, R., Turner, N. L., Walker, E. Y., Wang, T., Williams, G., Williams, S., Willie, K., Willie, R., Wong, W., Wu, J., Xu, C., Yang, R., Yatsenko, D., Ye, F., Yin, W., and Yu, S.-c. (2021). Functional connectomics spanning multiple areas of mouse visual cortex. Technical report, bioRxiv. Section: New Results Type: article.
- Motta, A., Schurr, M., Staffler, B., and Helmstaedter, M. (2019). Big data in nanoscale connectomics, and the greed for training labels. *Current Opinion in Neurobiology*, 55:180–187.
- Staffler, B. S. (2020). Machine Learning for Connectomics. page 126.

Turner, N. L., Macrina, T., Bae, J. A., Yang, R., Wilson, A. M., Schneider-Mizell, C., Lee, K., Lu, R., Wu, J., Bodor, A. L., Bleckert, A. A., Brittain, D., Froudarakis, E., Dorkenwald, S., Collman, F., Kemnitz, N., Ih, D., Silversmith, W. M., Zung, J., Zlateski, A., Tartavull, I., Yu, S.-c., Popovych, S., Mu, S., Wong, W., Jordan, C. S., Castro, M., Buchanan, J., Bumbarger, D. J., Takeno, M., Torres, R., Mahalingam, G., Elabbady, L., Li, Y., Cobos, E., Zhou, P., Suckow, S., Becker, L., Paninski, L., Polleux, F., Reimer, J., Tolias, A. S., Reid, R. C., Costa, N. M. d., and Seung, H. S. (2020). Multiscale and multimodal reconstruction of cortical structure and function. *bioRxiv*, page 2020.10.14.338681. Publisher: Cold Spring Harbor Laboratory Section: New Results.

Urakubo, H., Bullmann, T., Kubota, Y., Oba, S., and Ishii, S. (2019). UNI-EM: An Environment for Deep Neural Network- Based Automated Segmentation of Neuronal Electron Microscopic Images. *Scientific Reports*, 9(1):19413. Number: 1 Publisher: Nature Publishing Group.

Wolf, S. (2020). *Machine Learning for Instance Segmentation*. page 135.

Wolny, A., Cerrone, L., Vijayan, A., Tofanelli, R., Barro, A. V., Louveaux, M., Wenzl, C., Strauss, S., Wilson-Sánchez, D., Lymbouridou, R., Steigleder, S. S., Pape, C., Bailoni, A., Duran-Nebreda, S., Bassel, G. W., Lohmann, J. U., Tsiantis, M., Hamprecht, F. A., Schneitz, K., Maizel, A., and Kreshuk, A. (2020). Accurate and versatile 3D segmentation of plant tissues at cellular resolution. *eLife*, 9:e57613. Publisher: eLife Sciences Publications, Ltd.

Zhao, T., Olbris, D. J., Yu, Y., and Plaza, S. M. (2018). NeuTu: Software for Collaborative, Large-Scale, Segmentation-Based Connectome Reconstruction. *Frontiers in Neural Circuits*, 12.

Zhuang, W., Tristan, H., Takashima, R., Takiguchi, T., and Ariki, Y. (2020). Optimizing the Computational Efficiency of 3D Segmentation Models for Connectomics. Number: 2500 Publisher: EasyChair.

Decision Letter, first revision:

Dear Dr. Funke,

Thank you for submitting your revised manuscript "Local Shape Descriptors for Neuron Segmentation" (NMETH-BC46575A). It has now been seen by three of the original referees and their comments are below. The reviewers find that the paper has improved in revision, and therefore we'll be happy in principle to publish it in *Nature Methods*, pending minor revisions to satisfy the referees' final requests and to comply with our editorial and formatting guidelines.

TRANSPARENT PEER REVIEW

Nature Methods offers a transparent peer review option for new original research manuscripts submitted from 17th February 2021. We encourage increased transparency in peer review by publishing the reviewer comments, author rebuttal letters and editorial decision letters if the authors agree. Such peer review material is made available as a supplementary peer review file. Please state in the cover letter 'I wish to participate in transparent peer review' if you want to opt in, or 'I do not wish to participate in transparent peer review' if you don't. Failure to state your preference will result in delays in accepting your manuscript for publication.

Thank you again for your interest in Nature Methods Please do not hesitate to contact me if you have any questions.

Best regards,
Nina

Nina Vogt, PhD
Senior Editor
Nature Methods

ORCID

Reviewer #1 (Remarks to the Author):

I have no more comments.

Reviewer #3 (Remarks to the Author):

I am satisfied with the authors comments and additional experiments. LSD will certainly be a useful tool for segmenting large volume EM datasets.

Reviewer #4 (Remarks to the Author):

After reviewing the manuscript, the other reviewer responses, and the response to the reviewers I would recommend the paper for acceptance, under the condition that the authors work with the editor to incorporate the ssTEM mouse cortex results into the main text in an appropriate fashion (see major concern 1).

Major concerns:

The authors appear to have done the experiments to analyze the performance of their approach on TEM based mouse data, which addresses the combination of concerns raised by multiple reviewers. However, these results are not integrated into the text of the paper. In fact, there are no references to those experiments in the main text. This leaves most readers with the same questions the reviewers had and is a disservice to the paper. Further, there are statements which remain in the paper which directly imply such experiments have not been done: "Finally, the SNEMI3D dataset has an anisotropy factor of ~ 5 , whereas the data we test on here has an anisotropy factor of either ~ 2 (Zebrafinch) or is isotropic (Hemi-brain, Fib-25)." This leaves the reader thinking that they have not tested the approach on more anisotropic data, when they have in fact done so. I understand that the authors cannot afford nor have the expertise to train, run and optimize the FFN approach on anisotropic TEM data from mouse cortex, which makes it awkward to incorporate into the text in the same way as the other datasets. However, in a way this reinforces a key point they are trying to make (that the cost of FFNs make them generally not accessible) and I think can be easily addressed with some modest rewriting of the text.

Minor comments:

1. For the record, I disagree with the statement of reviewer 2. "The experimental results indicate that the proposed methods still overall underperforms FFN, although having higher throughput. This limits the practical significance of the proposed methods in computing neuron segmentation, where the segmentation quality is of higher priority."

Computational cost is a key barrier to advancement in the field of connectomics. The baseline method described in this paper currently costs hundreds of thousands of dollars per mm^3 . FFNs are likely a multi-million dollar scale per mm^3 , therefore making them practically impossible for most groups to run at this scale. Scientists have a fixed budget, which they can invest in segmentation and in proofreading, it's the total cost to achieve quality sufficient to address a scientific question which must

be optimized. I also think the authors make a compelling case for how ERL is biased against false mergers, which is not of practical import in considering total proofreading time. Insisting on ERL as the right metric and that computational cost doesn't matter is not grounded in the practicalities of accomplishing science in the field.

I believe Nature Methods should be publishing computational techniques which are of comparable quality but much lower cost, just as they do for experimental techniques. For example, most recent advancements in single cell sequencing (drop-seq) or mFISH methods (barSEQ, MERFISH) have not been higher data quality, but lower cost and increased throughput. Computational approaches that achieve equivalent results or close approximations that get very close in much less runtime have a long history of being of practical impact in computational science. The Fast Fourier Transform being the most obvious exact example.

2. The authors state in the response to reviewers

“In our manuscript, we intend to keep the post-processing the same to evaluate neuron segmentation methods under identical conditions. “ and

“In our manuscript, however, we compare different methods for one step of this pipeline, i.e., the impact of the neural network that predicts a neuron segmentation. There are many other variables outside of neural networks that contribute to the accuracy of the whole pipeline (e.g., alignment, masking, and post-processing). Therefore, the amount of engineering put into specific datasets will greatly influence the performance of differing pipelines.”

However, they also state in the paper “We further include a comparison against FFN segmentations, which were made available to us by the authors of Januszewski et al. (2018)” This network has a very different approach to segmentation, and so its pipeline is not comparable, and the philosophy of keeping everything other than affinity prediction the same is not possible. To that end I don't agree with the first quote, as “neuron segmentation” is more than just affinity prediction. I do agree with the second quote above, but it's not clear to me that the paper is consistently applying this standard. The authors should be more explicit about this, else people will assume that equal dataset specific engineering went into each method evaluated here when in fact did not.

I would suggest that the authors should include a statement in section 2.3 in this vein. Including the fact that various pipeline variations have been proposed and might interact with the affinity prediction method, whereas the FFN segmentation was evaluated as given and what dataset specific optimizations were done is not clear, as a standardized comparison was not possible.

3. The authors should comment on the reasons for the sizeable difference in the ratio of the terraFLOP cost and runtime cost per μm^3 between FFNs and the convolutional networks. The FFN network is only 12.2-15x less efficient from the runtime perspective, but ~ 80 -161x less efficient with the terraFLOP calculation. Also, they should comment explicitly on the differences in GPU hardware used and how that is likely to affect the metrics, namely that the V100 they used is roughly 2x more powerful than the P100 the FFN segmentation was performed on. This might fit best in the future directions section which could discuss how changes to different pipelines and implementation details might affect future throughput. In that same vein, they should comment on the coarse-to-fine approach described in the supplement of the FFN paper which reported a 5x improvement in throughput and suggests how multi-resolution methods could further improve throughput of different segmentation approaches.

Author Rebuttal, first revision:

We thank the reviewers and the editor for their helpful comments on our manuscript.

The main points brought up in the final review concerned the addition of the ssTEM results to the main text, an in-depth discussion of dataset-specific engineering, and computational cost discrepancies. The following sections address individual points brought up in the reviews, following the same order. The original reviews are in-lined and framed for visual distinction. Changes to the manuscript and the Supplemental Material are underlined and contain a pointer to the corresponding section in this document.

REVIEWER 1

I have no more comments.

REVIEWER 2

No comments.

REVIEWER 3

I am satisfied with the authors comments and additional experiments. LSD will certainly be a useful tool for segmenting large volume EM datasets.

REVIEWER 4

After reviewing the manuscript, the other reviewer responses, and the response to the reviewers I would recommend the paper for acceptance, under the condition that the authors work with the editor to incorporate the ssTEM mouse cortex results into the main text in an appropriate fashion (see major concern 1).

Major issues:

Inclusion of ssTEM results in main text

The authors appear to have done the experiments to analyze the performance of their approach on TEM based mouse data, which addresses the combination of concerns raised by multiple reviewers. However, these results are not integrated into the text of the paper. In fact, there are no references to those experiments in the main text. This leaves most readers with the same questions the reviewers had and is a disservice to the paper. Further, there are statements which remain in the paper which directly imply such experiments have not been done: "Finally, the SNEMI3D dataset has an anisotropy factor of ~ 5 , whereas the data we test on here has an anisotropy factor of either ~ 2 (Zebrafinch) or is isotropic (Hemi-brain, Fib-25)."; This leaves the reader thinking that they have not tested the approach on more anisotropic data, when they have in fact done so. I understand that the authors cannot afford nor have the expertise to train, run and optimize the FFN approach on anisotropic TEM data from mouse cortex, which makes it awkward to incorporate into the text in the same way as the other datasets. However, in a way this reinforces a key point they are trying to make (that the cost of FFNs make them

generally not accessible) and I think can be easily addressed with some modest rewriting of the text.

We have revised the results section to include the ssTEM results.

Minor issues:

Discuss dataset specific engineering

The authors state in the response to reviewers "In our manuscript, we intend to keep the post-processing the same to evaluate neuron segmentation methods under identical conditions.", and "In our manuscript, however, we compare different methods for one step of this pipeline, i.e., the impact of the neural network that predicts a neuron segmentation. There are many other variables outside of neural networks that contribute to the accuracy of the whole pipeline (e.g., alignment, masking, and post-processing). Therefore, the amount of engineering put into specific datasets will greatly influence the performance of differing pipelines."

However, they also state in the paper "We further include a comparison against FFN segmentations, which were made available to us by the authors of Januszewski et al. (2018)". This network has a very different approach to segmentation, and so its pipeline is not comparable, and the philosophy of keeping everything other than affinity prediction the same is not possible. To that end I don't agree with the first quote, as "neuron segmentation" is more than just affinity prediction. I do agree with the second quote above, but it's not clear to me that the paper is consistently applying this standard. The authors should be more explicit about this, else people will assume that equal dataset specific engineering went into each method evaluated here when it in fact did not.

I would suggest that the authors should include a statement in section 2.3 in this vein. Including the fact that various pipeline variations have been proposed and might interact with the affinity prediction method, whereas the FFN segmentation was evaluated as given and what dataset specific optimizations were done is not clear, as a standardized comparison was not possible.

We have included a statement discussing this at the end of the methods post-processing section.

Discuss computational costs

The authors should comment on the reasons for the sizeable difference in the ratio of the terraFLOP cost and runtime cost per μm^3 between FFNs and the convolutional networks. The FFN network is only 12.2-15x less efficient from the runtime perspective, but ~ 80 -161x less efficient with the terraFLOP calculation. Also, they should comment explicitly on the differences in GPU hardware used and how that is likely to affect the metrics, namely that the V100 they used is roughly 2x more powerful than the P100 the FFN segmentation was performed on. This might fit best in the future directions section which could discuss how changes to different pipelines

and implementation details might affect future throughput. In that same vein, they should comment on the coarse-to-fine approach described in the supplement of the FFN paper which reported a 5x improvement in throughput and suggests how multi-resolution methods could further improve throughput of different segmentation approaches.

The difference between runtime and teraFLOPs between affinity-based methods and FFN is most likely to do with data input/output (I/O). The total processing time for affinity-based methods includes data loading and normalization on CPU before transfer to GPU. These numbers are also highly dependent on factors specific to a given compute cluster which are hard to control for (such as file systems, networking, etc). These factors can be further engineered to increase performance. We have added a statement about runtime in Supplementary Section F. We agree that future work could leverage multi-resolution approaches. We have added a comment about multi-resolution methods at the end of the throughput section in the main text.

Final Decision Letter:

Dear Dr Funke,

I am pleased to inform you that your Article, "Local Shape Descriptors for Neuron Segmentation", has now been accepted for publication in Nature Methods. Your paper is tentatively scheduled for publication in our January print issue, and will be published online prior to that. The received and accepted dates will be July 13th, 2021 and November 1st, 2022. This note is intended to let you know what to expect from us over the next month or so, and to let you know where to address any further questions.

Over the next few weeks, your paper will be copyedited to ensure that it conforms to Nature Methods style. Once your paper is typeset, you will receive an email with a link to choose the appropriate publishing options for your paper and our Author Services team will be in touch regarding any additional information that may be required.

Your paper will now be copyedited to ensure that it conforms to Nature Methods style. Once proofs are generated, they will be sent to you electronically and you will be asked to send a corrected version

within 24 hours. It is extremely important that you let us know now whether you will be difficult to contact over the next month. If this is the case, we ask that you send us the contact information (email, phone and fax) of someone who will be able to check the proofs and deal with any last-minute problems.

If, when you receive your proof, you cannot meet the deadline, please inform us at rjsproduction@springernature.com immediately.

Once your manuscript is typeset and you have completed the appropriate grant of rights, you will receive a link to your electronic proof via email with a request to make any corrections within 48 hours. If, when you receive your proof, you cannot meet this deadline, please inform us at rjsproduction@springernature.com immediately.

Once your paper has been scheduled for online publication, the Nature press office will be in touch to confirm the details.

Content is published online weekly on Mondays and Thursdays, and the embargo is set at 16:00 London time (GMT)/11:00 am US Eastern time (EST) on the day of publication. If you need to know the exact publication date or when the news embargo will be lifted, please contact our press office after you have submitted your proof corrections. Now is the time to inform your Public Relations or Press Office about your paper, as they might be interested in promoting its publication. This will allow them time to prepare an accurate and satisfactory press release. Include your manuscript tracking number NMETH-A46575B and the name of the journal, which they will need when they contact our office.

About one week before your paper is published online, we shall be distributing a press release to news organizations worldwide, which may include details of your work. We are happy for your institution or funding agency to prepare its own press release, but it must mention the embargo date and Nature Methods. Our Press Office will contact you closer to the time of publication, but if you or your Press Office have any inquiries in the meantime, please contact press@nature.com.

Please note that *Nature Methods* is a Transformative Journal (TJ). Authors may publish their research with us through the traditional subscription access route or make their paper immediately open access through payment of an article-processing charge (APC). Authors will not be required to make a final decision about access to their article until it has been accepted. [Find out more about Transformative Journals](https://www.springernature.com/gp/open-research/transformative-journals)

To assist our authors in disseminating their research to the broader community, our SharedIt initiative provides you with a unique shareable link that will allow anyone (with or without a subscription) to read the published article. Recipients of the link with a subscription will also be able to download and print the PDF. As soon as your article is published, you will receive an automated email with your shareable link.

Please note that you and your coauthors may order reprints and single copies of the issue containing your article through Nature Portfolio's reprint website, which is located at <http://www.nature.com/reprints/author-reprints.html>. If there are any questions about reprints please send an email to author-reprints@nature.com and someone will assist you.

Best regards,
Nina

Nina Vogt, PhD
Senior Editor
Nature Methods